# A Smooth Sea Never Made a Skilled `SAILOR`:
# Robust Imitation via Learning to Search

**Arnav Kumar Jain**[*]
Mila- Quebec AI Institute
Université de Montréal

**Vibhakar Mohta**[*]
Carnegie Mellon University

**Subin Kim**
Cornell University

**Atiksh Bhardwaj**
Cornell University

**Juntao Ren**
Cornell University

**Yunhai Feng**
Cornell University

**Sanjiban Choudhury**
Cornell University

**Gokul Swamy**
Carnegie Mellon University

## Abstract

The fundamental limitation of the behavioral cloning (BC) approach to imitation learning is that it only teaches an agent what the expert did at states the expert visited. This means that when a BC agent makes a mistake which takes them out of the support of the demonstrations, they often don't know how to recover from it. In this sense, BC is akin to *giving the agent the fish* – giving them dense supervision across a narrow set of states – rather than teaching them *to fish*: to be able to reason independently about achieving the expert's outcome even when faced with unseen situations at test-time. In response, we explore *learning to search* (L2S) from expert demonstrations, i.e. learning the components required to, at test time, plan to match expert outcomes, even after making a mistake. These include *(1)* a world model and *(2)* a reward model. We carefully ablate the set of algorithmic and design decisions required to combine these and other components for stable and sample/interaction-efficient learning of recovery behavior without additional human corrections. Across a dozen visual manipulation tasks from three benchmarks, our approach `SAILOR` consistently out-performs state-of-the-art Diffusion Policies trained via BC on the same data. Furthermore, scaling up the amount of demonstrations used for BC by 5-10× still leaves a performance gap. We find that `SAILOR` can identify nuanced failures and is robust to reward hacking. Our code is available at `https://github.com/arnavkj1995/SAILOR`.

## 1 Introduction

The workhorse of modern imitation learning (IL) is behavioral cloning (BC, Pomerleau [1988]). From training Diffusion Policies (DPs, Chi et al. [2023]) on per-task expert demonstrations collected via a variety of teleoperation interfaces [Zhao et al., 2023, Chi et al., 2024, Wu et al., 2024] to Visual-Language-Action models (VLAs, Team et al. [2024], Kim et al. [2024], Intelligence et al. [2025]) trained on wider, multi-task datasets [Khazatsky et al., 2024, ONeill et al., 2024], we see the same *recipe* applied: collecting more data to train more expressive policy models. The latent hope here is that scaling will eventually lead to a "ChatGPT moment" for robotics [Vemprala et al., 2023].

---

[*]Equal Contribution.
Correspondence to Arnav <arnav-kumar.jain@mila.quebec> and Gokul <gswamy@cmu.edu>.

39th Conference on Neural Information Processing Systems (NeurIPS 2025).

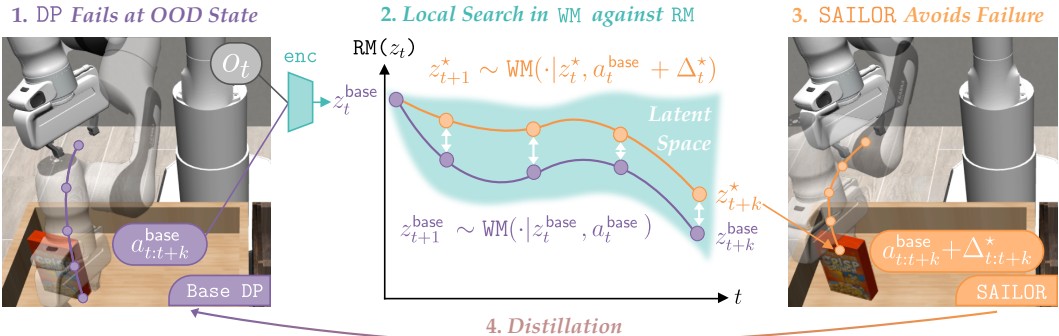

Figure 1: We introduce SAILOR, a method for *learning to search* from expert demonstrations. By learning world and reward models on a mixture of expert and base-policy data, we endow the agent with the ability to, at test time, reason about how to recover from mistakes that the base policy makes.

However, even for the *simpler* problem of language modeling where one doesn't have to deal with the complexities of embodiment (e.g. grounding, stochastic dynamics, partial observability, safety), simply scaling *next token prediction* (i.e. BC) was insufficient: we needed *interactive learning* in the form of Reinforcement Learning from Human Feedback (RLHF, Stiennon et al. [2020], Ouyang et al. [2022]) and more recently, Test-Time Scaling (TTS, Jaech et al. [2024], Guo et al. [2025]), to build robust systems. If internet-scale offline pretraining was insufficient to solve language modeling, it stands to reason that we'll need similar interactive learning algorithms to train (embodied) agents.

At heart, this is because even when they are trained on large amounts of data and over expressive policy classes, agents still sometimes make mistakes that take them out of the support of the offline data. This is a fundamental property of sequential decision-making: one has to deal with the consequences of their own prior actions. When faced with the resulting unseen situation, we'd like our agents to attempt to *recover* and match the expert's *outcome* (whenever it is possible to do so). The most direct approach to teach an agent this recovery behavior is to ask a human-in-the-loop to correct mistakes [Ross et al., 2011, Kelly et al., 2019, Spencer et al., 2020]. While simple, such an approach can be difficult to scale as it fundamentally makes *human* time the bottleneck for *robot* learning.

In an ideal world, we'd like our robots to be able to learn to recover from their *own* mistakes without additional human feedback. There are two fundamental capabilities an agent needs to reason at test-time about recovering from mistakes. The first is *prediction*: to understand the consequences of their proposed actions. The second is *evaluation*: to know which outcomes are preferable to others.

We propose an algorithmic paradigm that allows us to acquire both of these capabilities without requiring any sources of human data beyond the standard imitation learning pipeline. In other words, *a better recipe with the same ingredients*. In particular, rather than merely learning a policy from expert demonstrations, we propose learning a *local world model* (WM) and a *reward model* (RM) from demonstration and base policy data [Ren et al., 2024b]. By combining these components with a planning algorithm, we have the capability to, at test time, reason about how to recover from mistakes that the base policy makes by planning against our learned RM inside our learned WM. Thus, rather than mere imitation, we're *learning to search* (L2S, Ratliff et al. [2009]) from expert demonstrations. Our key insight is that *we can infer the latent search process required to recover from local mistakes without any more human feedback (e.g., corrections) than a standard behavioral cloning pipeline*.

Put differently, in contrast to approaches like BC and DAgger that *give the agent the fish* – i.e. relying on a human teacher to demonstrate desired or recovery behavior, we focus on teaching the agent *to fish*: *to develop the reasoning process required to match the expert's outcomes, even when faced with situations unseen in the training dataset*, often as a result of the agent's own earlier mistakes.

In our work, we focus on long-horizon visual manipulation tasks and therefore instantiate the L2S paradigm by using base Diffusion Policies [Chi et al., 2023], Dreamer World Models [Hafner et al., 2024], and the Model-Predictive Path Integral Control (MPPI, Williams et al. [2017]) Planner. Concretely, we learn a residual planner [Silver et al., 2018] that performs a *local search* at test time

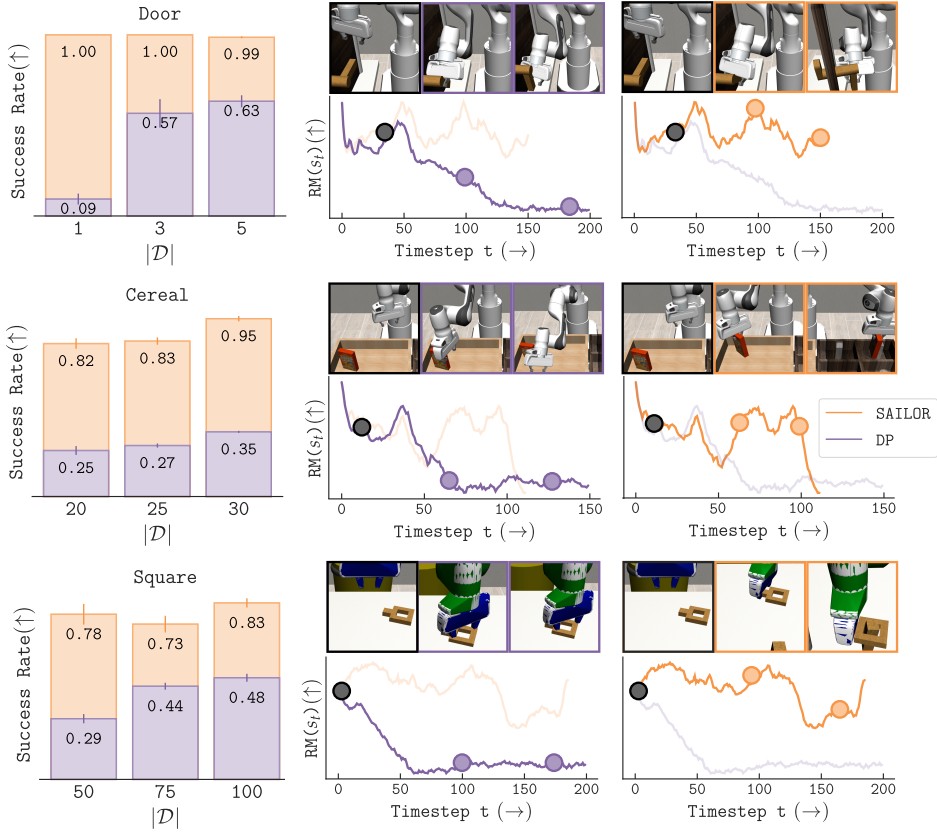

Figure 2: **Left:** we see `SAILOR` consistently out-perform diffusion policies trained on the same demos across various visual manipulation problems at multiple dataset scales $|\mathcal{D}|$. **Right:** `SAILOR`'s learned reward model is able to detect shared prefixes (black dots and frames), base policy failures (purple dots, lines, and frames) and `SAILOR`'s successes (orange dots, lines, and frames).

to correct mistakes the base policy makes. We call our composite architecture `SAILOR`: **Searching Across Imagined Latents Online for Recovery**. More specifically, our contributions are three-fold:

**1. We demonstrate that across a dozen visual manipulation problems at three different dataset scales, `SAILOR` outperforms Diffusion Policies. Simply scaling up the number of demonstrations used for DP by $\approx$ 5-10$\times$ still leaves a performance gap.** Furthermore, `SAILOR` is also significantly more interaction-efficient than more traditional model-free inverse RL methods that use state of the art Diffusion Policy RL algorithms like DPPO [Ren et al., 2024a] to directly update policy parameters.

**2. We carefully ablate the algorithmic and design decisions required to learn and combine the above components for stable and sample-efficient learning.** Specifically, we find that "warm starting" the hybrid world model training period, doing online hybrid world model fine-tuning [Ross and Bagnell, 2012, Vemula et al., 2023, Ren et al., 2024b], and periodically distilling the learned search algorithm into the base policy (akin to expert iteration [Anthony et al., 2017, Sun et al., 2018] or Guided Policy Search [Levine and Koltun, 2013]) are critical for performance.

**3. We investigate the fidelity and test-time scaling properties of our learned search algorithm.** We find that our learned reward model is able to identify nuanced failures that occur at different stages of our long-horizon manipulation tasks and that our composite stack is robust to reward hacking.

## 2 Related Work

**Imitation Learning.** The simplest approach to imitation learning is *behavioral cloning* (BC, Pomerleau [1988], Chi et al. [2023]), where one learns a policy via maximizing the likelihood of expert

actions at expert states. However, whether it is due to limited demonstrations [Swamy et al., 2022b], optimization error [Swamy et al., 2021], misspecification [Espinosa-Dice et al., 2025], or partial observability [Swamy et al., 2022a], a policy trained via BC will often make a mistake that takes it out of the support of the expert demonstrations, where it can continue to make errors, an issue known as *compounding errors* [Ross et al., 2011], which is unavoidable in general [Swamy et al., 2021].

Under the hood, the reason a BC policy makes mistakes is due to the *covariate shift* between the training input distribution (expert states) and the testing input distribution (learner states). Interacting with the environment allows us to generate samples from this test distribution. If we're able to further query the demonstrator in the loop, we can ask them for action labels at these states [Ross et al., 2011, Kelly et al., 2019, Spencer et al., 2020]. However, such approaches are often labor-intensive.

**Learning to Search = RM + WM.** In Learning to Search (L2S, Ratliff et al. [2009]), one learns the components (i.e., RM and a WM) required to, at test time, search for actions, rather than directly learning a policy. If there are no meaningful dynamics or stochasticity (e.g., append-only, auto-regressive language generation), one can eschew the WM and plan over the entire horizon via repeated sampling and scoring under the RM, an approach known as Best-of-N (BoN, Brown et al. [2024]). Otherwise, one samples some plans, performs stochastic rollouts inside the WM, calling the RM along the way to estimate performance before updating the sampling distribution. We use the `MPPI` algorithm of Williams et al. [2017] as our search procedure due to the strong performance it has demonstrated when deployed inside a learned WM on robotics problems [Hansen et al., 2022, 2024]. In contrast to prior L2S approaches. `SAILOR` learns from expert demonstrations alone (i.e., without test-time access to a ground-truth simulator as in Silver et al. [2017], Brown and Sandholm [2019] or *any* information about the ground-truth reward function as in Hansen et al. [2022, 2024]). [2]

After expending computation at test-time for a search procedure, it is often valuable to distill the search process back into the base policy, an approach known as *expert iteration* or *dual policy iteration* [Anthony et al., 2017, Sun et al., 2018]. While such approaches have long demonstrated strong performance in continuous control [Levine and Koltun, 2013, Wang et al., 2025], they have attracted renewed interest in the context of LLMs [Zelikman et al., 2022, Gandhi et al., 2024, Hosseini et al., 2024, Jain et al., 2025a]. `SAILOR` can be seen as a generalization of these ideas to continuous control problems with stochastic dynamics, visual observations, and unknown reward functions, which none of the above can handle directly. We ablate the value of ExIt-like updates and find they provide nontrivial improvement in policy performance. Recent work by Wu et al. [2025b] applies similar ideas on real-world visual manipulation problems but focuses on mode selection rather than more general behavior correction. `SAILOR` can be thought of fusing the paradigms of residual RL [Silver et al., 2018, Yuan et al., 2025, Ankile et al., 2024] and L2S by *learning to search for residuals*.

## 3 Robust Imitation via *Learning to Search*

We adopt the framework of a Partially Observed Markov Decision Process (POMDP, Kaelbling et al. [1998]) and use $\mathcal{O}$, $\mathcal{A}$, and $\mathcal{Z}$ to denote the observation, action, and latent spaces, respectively. Also, we use $\gamma \in [0, 1]$ to denote the discount factor and $k$ to denote our policy's planning horizon.

### 3.1 What is Learning to Search?

In `SAILOR`, we *learn a search algorithm* that generates residual plans $\Delta_{t:t+k}^{\star}$ to correct a nominal plan $a_{t:t+k}^{\mathsf{base}}$ generated by the base policy. Concretely, after learning a world model `WM`, reward model `RM`, and critic `V` from a combination of expert demonstrations and base policy rollouts, we perform repeated stochastic rollouts of potential corrected plans inside our `WM`, scoring the latent states $z_{t:t+k}$ with R and V, before selecting the plan with the highest estimated score. We then execute the first step of the corrected plan in the real world before re-planning in the style of *model predictive control*

---

[2]We note briefly that the term "learning to search" is also used to describe a class of methods for *structured prediction* problems, which encompass many natural language processing tasks [Chang et al., 2015, 2023]. However, these methods usually assume access to a queryable expert policy in the vein of DAgger [Ross et al., 2011] and AggraVaTe [Ross and Bagnell, 2014]. In contrast, we make no such assumptions, and instead focus on how best to give the learner the ability to search independently at test time without additional expert guidance.

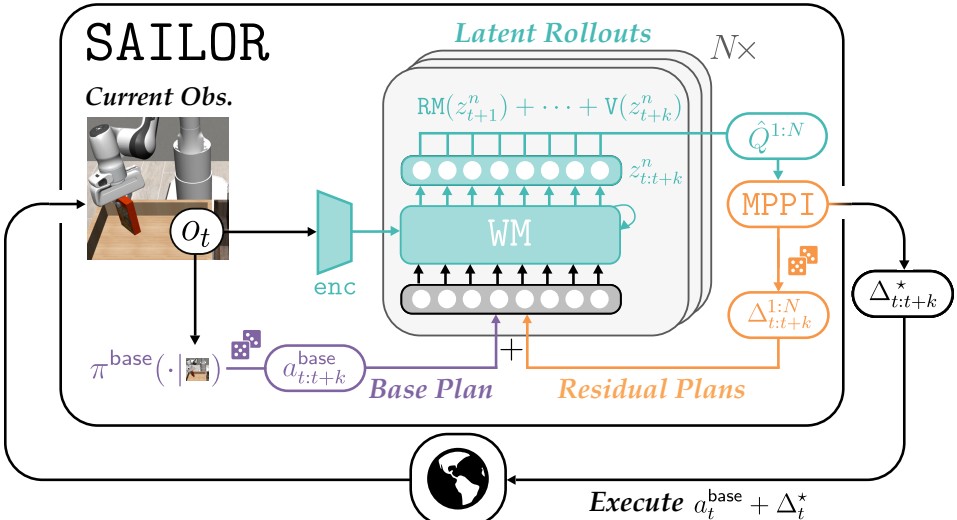

Figure 3: At inference time, SAILOR performs a search for residual plans to correct mistakes in the base policy's nominal plan in the latent world model WM against the learned reward model RM and critic V. It then executes the first step of the best corrected plan before re-planning, MPC-style.

---

**Algorithm 1** SAILOR (Inference)

1: **Input:** Base Policy $\pi^{\mathsf{base}}$, World Model WM, Reward Model RM, Critic Network V, Obs. $o_t$,
2: Sample nominal plan from base: $a^{\mathsf{base}}_{t:t+k} \sim \pi^{\mathsf{base}}(o_t)$.
3: **for** iteration $j$ in $1 \ldots J$ **do**
4:    // Can perform in parallel
5:    **for** residual plan $n$ in $1 \ldots N$ **do**
6:       Sample $\Delta^n_{t:t+k} \sim \mathcal{N}(\mu^j, \sigma^j)$.
7:       Execute plan in WM: $z^n_{t:t+k} \sim \mathrm{WM}(o_t, a^{\mathsf{base}}_{t:t+k} + \Delta^n_{t:t+k})$.
8:       Compute $\hat{Q}$s: $\hat{Q}^n \leftarrow \sum_{h=0}^{k-1} \gamma^h \mathrm{RM}(z^n_{t+h}) + \gamma^k \mathrm{V}(z^n_{t+k})$.
9:    **end for**
10:   // Update mean and std
11:   $\mu^{j+1}, \sigma^{j+1} \leftarrow \mathrm{MPPI\_update}(\mu^j, \sigma^j, \hat{Q}^{1:N}, \Delta^{1:N}_{t:t+k})$.
12: **end for**
13: **Return** corrected plan $a^\star_{t:t+k} \sim \mathcal{N}(a^{\mathsf{base}}_{t:t+k} + \mu^J, \sigma^J)$.

---

(MPC). More formally, at inference time, we attempt to solve the following *local search* problem:

$$\Delta^\star_{t:t+k} = \underset{\Delta_{t:t+k}}{\arg\max}\, \mathbb{E}_{z_t \sim \mathrm{WM}(z_{t-1}, a_{t-1})}\left[\sum_{h=0}^{k-1} \gamma^h \mathrm{RM}(z_{t+h}) + \gamma^k \mathrm{V}(z_{t+k}) \middle| o_t, a_{t:t+k} = a^{\mathsf{base}}_{t:t+k} + \Delta_{t:t+k}\right].$$

We then execute the first of these actions, $a^{\mathsf{base}}_t + \Delta^\star_t$, in the environment before re-planning on top of the fresh observation and base plan. We describe this process in Alg. 1 and visualize it in Fig. 3. We now describe each of these components before describing the phases of training. We use $\mathcal{D}$ to denote the set of expert demonstrations and $\mathcal{B}$ to denote a replay buffer of on-policy learner rollouts.

**Base Policy.** We assume access to a *base policy* $\pi^{\mathsf{base}}$ that can generate $k$-step plans given an observation $o_t$: $a^{\mathsf{base}}_{t:t+k} \sim \pi^{\mathsf{base}}(o_t)$. This could be an arbitrary policy pretrained on a wide set of data like a VLA [Team et al., 2024, Intelligence et al., 2025, Kim et al., 2024] or a task-specific Diffusion Policy (DP, Chi et al. [2023]) trained via behavioral cloning. We adopt the latter for our experiments for simplicity, but note that the general SAILOR framework does not require doing so. [3] As is standard practice [Chi et al., 2023], we use a ResNet-18 [He et al., 2016] encoder for our DP that takes in both image observations and the proprioceptive state of the robot and generates 8-step plans.

---

[3]Explicitly, we make no assumptions on the base policy, other than the ability to potentially fine-tune it via behavioral cloning (i.e., maximum likelihood estimation) for the *expert iteration* subroutine.

**Algorithm 2** SAILOR (Training)

---

1: **Input:** Base Policy $\pi^{\text{base}}$, Expert Demos $\mathcal{D}$
2: // Phase I: Warm Start
3: Collect on-policy rollouts: $\mathcal{B} \leftarrow \{(o_t, a_t) \sim \pi^{\text{base}}\}$.
4: // Co-train components on hybrid data
5: $\text{WM} \leftarrow \text{argmin}_{\text{WM}} \ell(\mathcal{D}, \mathcal{B}, \text{WM}), \text{RM} \leftarrow \text{argmin}_{\text{RM}} \ell(\mathcal{D}, \mathcal{B}, \text{RM}), \text{V} \leftarrow \text{argmin}_{\text{V}} \ell(\mathcal{D}, \mathcal{B}, \text{V})$.
6: // Phase II: Online Fine-Tuning
7: **for** iteration $j$ in $1 \ldots J$ **do**
8:     Collect on-policy rollouts: $\mathcal{B} \leftarrow \mathcal{B} \cup \{(o_t, a_t) \sim \text{SAILOR}\}$.
9:     $\text{WM} \leftarrow \text{argmin}_{\text{WM}} \ell(\mathcal{D}, \mathcal{B}, \text{WM}), \text{RM} \leftarrow \text{argmin}_{\text{RM}} \ell(\mathcal{D}, \mathcal{B}, \text{RM}), \text{V} \leftarrow \text{argmin}_{\text{V}} \ell(\mathcal{D}, \mathcal{B}, \text{V})$.
10:     // Phase III: Expert Iteration
11:     **if** $j \% m == 0$ **then**
12:         Relabel $\mathcal{B}^{\text{distill}} \leftarrow \{(o_t, \text{SAILOR}(o_t)) | o_t \in \mathcal{B}\}$.
13:         Distill $\pi^{\text{base}} = \text{argmin}_{\pi} \ell(\mathcal{B}^{\text{distill}}, \pi)$.
14:     **end if**
15: **end for**
16: **Return** $\pi^{\text{base}}$, WM, RM, V.

---

**World Model.** To avoid the complexity of modeling the dynamics of and planning directly in the space of high-dimensional observations, we adopt the Recurrent State-Space Model architecture (RSSM, Hafner et al. [2019, 2024]). This means our world model is composed of three key components, i.e. $\text{WM} = \{\text{enc}, \text{f}, \text{dec}\}$. The first, the *encoder* $\text{enc} : \mathcal{Z} \times \mathcal{O} \times \mathcal{A} \to \mathcal{Z}$ encodes the observation into latent space i.e. $z_t = \text{enc}(z_{t-1}, a_{t-1}, o_t)$. The second, the *latent dynamics model* $\text{f} : \mathcal{Z} \times \mathcal{A} \to \Delta(\mathcal{Z})$ predicts the next latent after taking an action, i.e. $z_t \approx \text{f}(z_{t-1}, a_{t-1})$. The third, the decoder $\text{dec} : \mathcal{Z} \to \mathcal{O}$ tries to reconstruct the observation from the latent state, i.e. $\text{dec}(z_t) \approx o_t$. In SAILOR, *we train all three of these components on hybrid data*, i.e. a mixture of $\mathcal{D}$ and $\mathcal{B}$ – see Appendix B for all loss functions. As argued by Ross and Bagnell [2012], Vemula et al. [2023], Ren et al. [2024b], while such a world model is only *locally accurate* on the expert's and learner's state distributions, a policy that looks good when evaluated in such a model is guaranteed to do well in the real world. [4]

**Reward Model.** Intuitively speaking, we train a reward model $\text{RM} : \mathcal{Z} \to \mathbb{R}$ to score the latent states of our world model by how "expert-like" they are, which allows us to plan to match expert outcomes in the imagined future. More formally, we train a *discriminator* between the latent embeddings of expert and learner rollouts using the moment-matching loss proposed by Swamy et al. [2021]:

$$\ell(\mathcal{D}, \mathcal{B}, \text{RM}) = \mathbb{E}_{(z,a,o)\sim\mathcal{B}}[\text{RM}(\text{enc}(z, a, o))] - \mathbb{E}_{(z,a,o)\sim\mathcal{D}}[\text{RM}(\text{enc}(z, a, o))]. \quad (1)$$

We also add in a gradient penalty Gulrajani et al. [2017] to the above to stabilize training. We iteratively update the RM over the course of training to ensure that we are able to detect mistakes made by the current iteration of the composite SAILOR stack, similar to inverse RL procedures [Ziebart et al., 2008, Ho and Ermon, 2016]. While we do not dwell on the theoretical implications thereof, we note in passing that minimizing the above across a set of potential reward functions corresponds to bounding the performance difference between the expert and the learner under any of these rewards, thereby avoiding compounding errors unlike purely offline behavioral cloning [Swamy et al., 2021].

**Critic.** To enable truncated horizon rollouts in our WM of $k$ steps, we learn a critic V that acts as a terminal cost estimate. The critic V is trained to predict bootstrapped $\lambda$-returns (Sutton et al. [1998]):

$$\mathbf{v}_t^{\lambda} = \text{RM}(z_t) + \gamma\Big((1-\lambda)\text{V}(z_t) + \lambda\mathbf{v}_{t+1}^{\lambda}\Big), \quad \mathbf{v}_{t+k}^{\lambda} = \text{V}(z_{t+k}). \quad (2)$$

In our experiments, we train an ensemble of 5 critic networks (Ball et al. [2023], Chen et al. [2021]). When computing terminal cost estimates, we take the mean of 2 randomly sampled critics and subtract an uncertainty penalty proportional to the standard deviation across the entire ensemble.

**The Three Phases of Training a Seaworthy SAILOR.** As outlined in Algorithm 2, training proceeds in three phases. In *Phase I: Warm-Start* (Lines 2-5), we perform rollouts with the base policy $\pi^{\text{base}}$

---

[4]While we do not investigate this in our experiments, the above theory still holds if the world model is trained on a wide data distribution that covers both the expert and learner's visitation distribution. Thus, a particularly interesting future direction is to train a powerful *foundation world model* [Agarwal et al., 2025, Bruce et al., 2024, Parker-Holder et al., 2024] on internet-scale robotics datasets [Khazatsky et al., 2024, ONeill et al., 2024], before plugging it into the SAILOR framework to allow for wider-ranging recovery from mistakes.

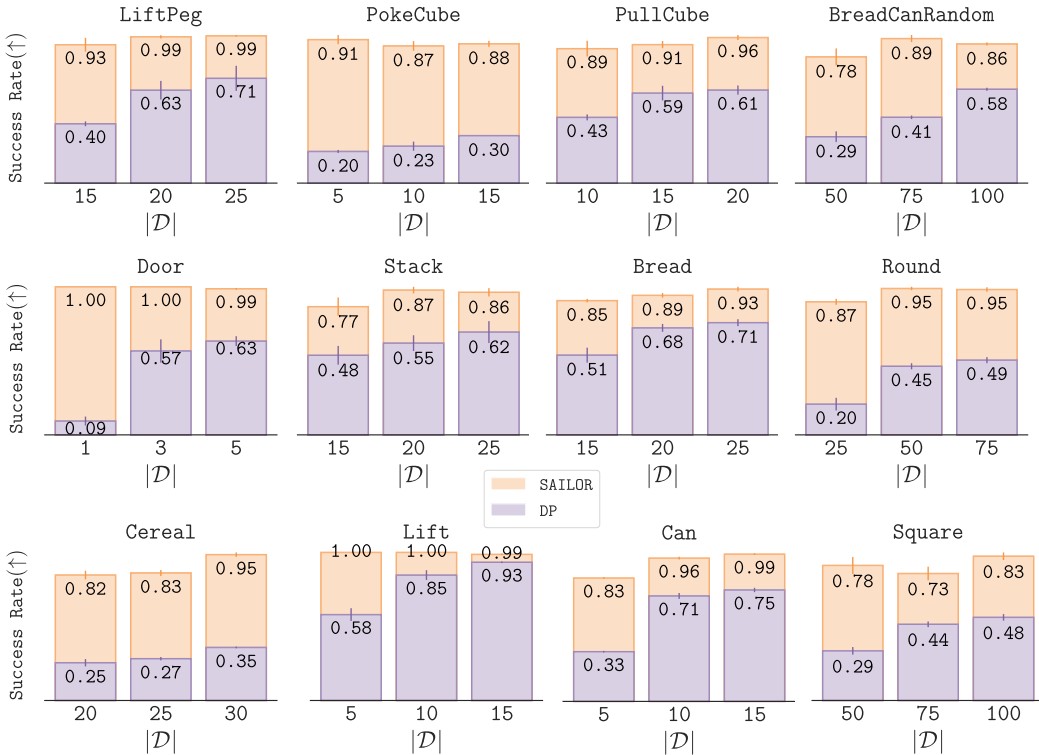

Figure 4: Across 12 visual manipulation problems from 3 benchmarks, SAILOR consistently outperforms diffusion policy (DP) trained on the same demos, where $|\mathcal{D}|$ denotes the number of demos.

to pre-fill the buffer $\mathcal{B}$, before co-training the world model WM, reward model RM, and critic V on a mixture of data from $\mathcal{B}$ and the expert demonstrations $\mathcal{D}$. In *Phase II: Online Fine-Tuning* (Lines 6-15), we instead perform rollouts with the entire SAILOR stack, periodically performing hybrid training of the WM, RM, and V. [5] In *Phase III: Expert Iteration* (Lines 11-14), we distill the outputs of test-time search into the base policy to avoid the need to expend compute if the learner ends up in a similar situation in the future. More formally, we take some of the most recent $(o, a)$ pairs in buffer $\mathcal{B}$, post-hoc relabel them by calling the SAILOR stack on the observation $o$ to generate new action labels, and fine-tune the base policy $\pi^{\text{base}}$ via behavioral cloning. This can be seen as using SAILOR as an expert for a DAgger [Ross et al., 2011] update of the base policy, recycling prior test-time compute.

## 4 Experiments

**Benchmarks.** We evaluate the efficacy of SAILOR on 12 challenging visual manipulation problems. This includes 3 tasks from Robomimic [Mandlekar et al., 2021b] {Lift, Can, Square}, 6 from RoboSuite [Zhu et al., 2020] {Door, Stack, Bread, Cereal, Round, BreadCanRandom}, and 3 from ManiSkill [Tao et al., 2025] {PullCube, PokeCube, LiftPeg}. These include diverse tasks: multi-step pick-and-place (BreadCanRandom), articulated object manipulation (Door), and tool use (Square, Round, PokeCube). For Robomimic tasks, we use the provided demonstrations, while we collect our own demonstrations using a SpaceMouse controller for the other suites. To solve each task, SAILOR is provided with a set of expert demonstrations $\mathcal{D}$, along with a budget specifying the maximum number of environment interactions it can perform. The observation space consists of RGB images from a wrist-mounted camera and a third-person camera mounted in front of the agent, and the proprioceptive states. More details are provided in App. D and E.

---

[5] One can consider training on buffer $\mathcal{B}$ as a variant of Follow the Regularized Leader [McMahan, 2011], a stable *no-regret algorithm*. More formally, by plugging in the bounds of Ross and Bagnell [2012] into the reduction of Ren et al. [2024b], one can derive performance bounds for SAILOR under standard assumptions.

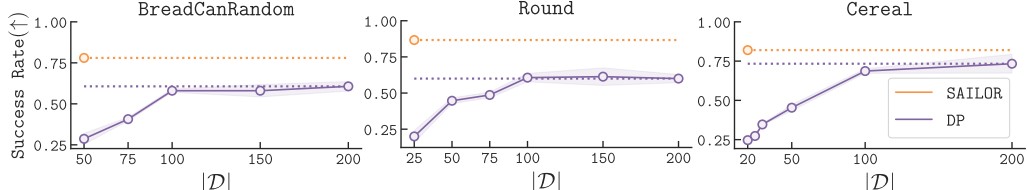

Figure 5: We see that simply scaling up the amount of demos $|\mathcal{D}|$ used for training DP via behavioral cloning by 5-10× often plateaus in performance and is unable to match the performance of SAILOR.

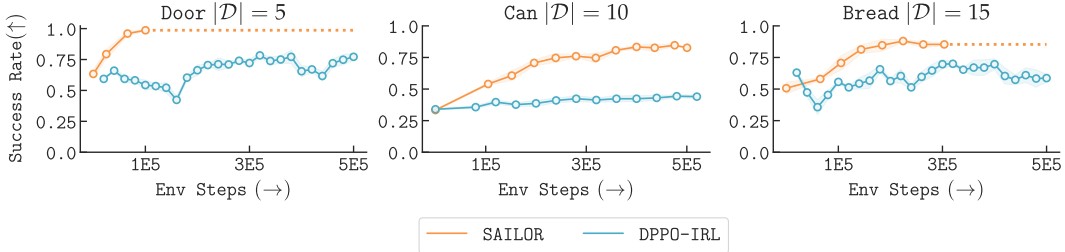

Figure 6: We find that SAILOR is significantly more interaction-efficient than a model-free inverse RL baseline that directly updates DP policy parameters via DPPO [Ren et al., 2024a].

**Algorithms.** We compare three imitation learning methods: Diffusion Policies (DP, Chi et al. [2023]) trained via BC, a model-free inverse RL method (DPPO-IRL) that directly updates DP parameters via RL (DPPO, Ren et al. [2024a]) against a learned RM with a separate encoder, and SAILOR. For evaluation, we measure the Success Rate (SR) across 50 rollouts, and report the mean and standard error obtained with 3 seeds. More details on the implementation of the methods are provided in App. B, C. All methods were trained on 1 NVIDIA 6000 Ada GPU with 48 GB of memory.

### 4.1 Results

**Can SAILOR outperform DP trained on the same $\mathcal{D}$?** Fig. 4 compares SAILOR and DP across various tasks and size of demonstration datasets. We observe that SAILOR significantly outperforms DP across all environments and dataset scales considered, indicating that the learning to search paradigm allows us to squeeze out significantly more from the same expert data. We find particularly large gaps in the low-data regime, reflecting how L2S inherits sample-efficiency benefits of inverse RL approaches that learn *verifiers* from human data rather than just learning policies / *generators* [Swamy et al., 2025].

**Does DP catch up with more data?** A natural question after viewing the preceding results might be as to whether more demonstrations could close the gap between DP and SAILOR, reflecting robot learning's current emphasis on large-scale data collection [ONeill et al., 2024, Khazatsky et al., 2024]. In Fig. 5, we observe that while DP improves with more expert data, the performance plateaus after 100 demonstrations. This means that when we scale up the number of provided demonstrations to 200, ≈ 10× the amount provided to SAILOR, DP is still unable to match our method's performance.

Note further that these are expert demonstrations collected directly for the target environment. One can imagine that as practitioners choose to scale up offline data with sources such as human-video demonstrations and Internet-scale pretraining, data which is by design even less in-distribution to the task at hand, we expect a pure BC model to exhibit diminishing performance gains. SAILOR, by contrast, might be able to absorb that same large, noisy dataset into its WM and RM. Pre-training these components on broad "notions of success" may yield robust dynamics priors and value estimates that drive on-policy action distillation and keep improving the policy long after BC has plateaued.

**How much real-world interaction does the WM save us?** Model-based approaches are well-known to be more interaction-efficient than their model-free analogs, a benefit SAILOR inherits. In particular, each observation isn't treated as a single data point. Instead, SAILOR uses it as a seed to spawn an entire tree of counter-factual trajectories, not only slashing the number of costly real trials needed

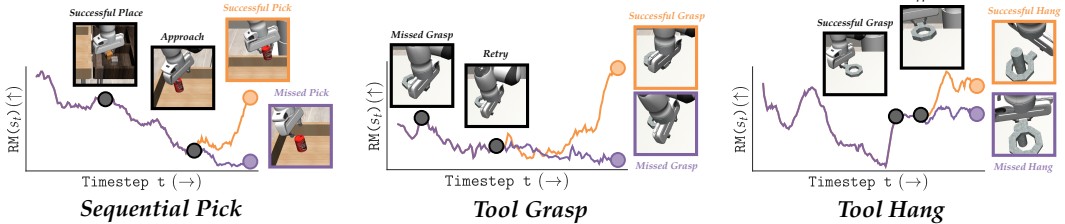

Sequential Pick     Tool Grasp     Tool Hang

Figure 7: At different stages of multi-step task execution, we see our learned reward model RM able to identify a variety of nuanced failures and SAILOR able to counterfactually avoid them. We generate these plots by rolling out the base policy until it fails at the given task (purple line), resetting the agent to a state where failure is not yet guaranteed, and letting SAILOR complete the rest of the episode (orange line). We use black dots to mark shared prefixes, purple dots to mark the base policy's failure suffixes, and orange dots to mark SAILOR's succesful suffixes.

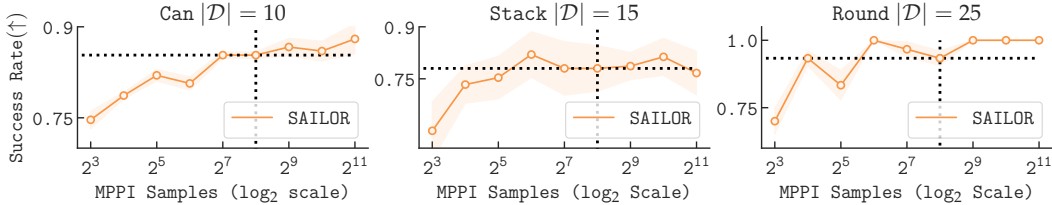

Figure 8: We use 256 samples (analogous to $N$ in BoN) for the MPPI planning process at train-time (black cross). We see that more compute leads to higher performance up to 256 and do not see a degradation in performance even with $8\times$ as many samples, indicating robustness to reward hacking.

to reach a given performance, but also pruning potentially high-variance or dangerous paths before executing them on the physical hardware. To quantify the size of this benefit, we compare SAILOR to a model-free IRL algorithm DPPO-IRL that directly updates policy parameters after performing rollouts in the real world. In Fig. 6, we consistently see that even with $5\times$ the interaction budget, DPPO-IRL is unable to match the performance of SAILOR, reflecting the importance of the WM.

**Can the RM detect nuanced failures?** We now explore qualitatively what our learned RM is detecting. We do this by sampling trajectories that fail to ultimately complete the task from the base DP, truncate at a prefix where failure is not yet guaranteed, and then roll out SAILOR counterfactually from these states to recover from mistakes. In Fig. 7, we see that our RM is able to detect nuanced failures that occur at various stages of a complex task (e.g. a narrowly missed tool hang after a successful grasp).

**How robust is SAILOR to reward hacking?** A common concern with test-time scaling approaches is that with enough compute, they may "hack" a learned proxy reward model and perform worse on the ground-truth metric [Gao et al., 2023]. We explore the test-time scaling properties of SAILOR by scaling up the number of samples generated in the MPPI planning procedure, which is analogous to the $N$ in BoN. We see in Fig. 12 that up to the number of samples used for training (256), more compute consistently leads to better performance. Furthermore, we do not see a degradation in performance even with $8\times$ as many samples, reflecting a high degree of robustness to reward hacking.

## 4.2 What Matters in Learning To Search?

We now ablate the importance of several parts of the overall SAILOR pipeline.

**Warm-Start.** We found that allocating $\approx 20\%$ of the interaction budget to a "warm-start phase" significantly boosts final performance, as shown in the leftmost part of Fig. 9. We attribute this to having the WM, RM, and V be accurate on $\pi^{\text{base}}$'s distribution, reducing exploration in Phase II. We find performance plateaus when using $> 20\%$ of the interaction budget for warm-starting (App. A.1).

**Hybrid WM Training.** Another component that has a significant effect on model performance is using a mix of expert and learner data to update the WM. We observe in Fig. 9, center, that this "hybrid" fitting of the WM not only has the potential to improve initial model performance (as for PokeCube), but can also improve final performance (as for Cereal), reflecting the insights of Ren et al. [2024b].

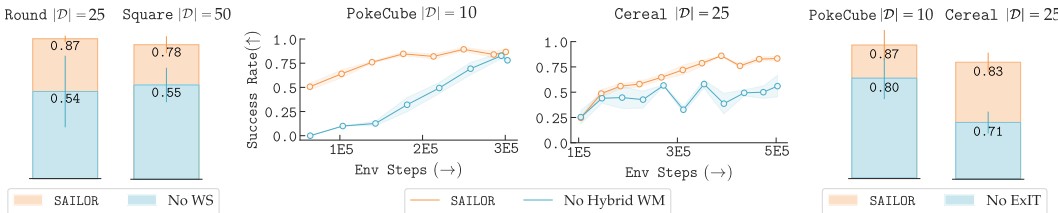

Figure 9: We ablate the impact of three components of our overall training pipeline: warm starting, hybrid world model training, and expert iteration, and find that all three improve performance.

**Expert Iteration.** In Fig. 9, right, we see that the inclusion of the expert iteration subroutine boosts model performance. This can be attributed to the base policy knowing how to recover from mistakes its prior iterations would have made, effectively recycling test-time compute into a new base policy.

**Residual Planning.** We ablate the importance of using test-time planning to search for residuals online instead of a precomputed residual policy (RP). Similar to the residual planner, the RP takes as input both the latent state and base policy action. It is trained via standard actor-critic techniques as in Dreamer [Hafner et al., 2024]. In Fig. 10, we see that across the board, explicit test-time planning out-performs a residual policy. As additional statistics, we also include numbers for the wall-clock time overhead of running MPPI, which we find to be somewhat limited. We also ablate the amount of MPPI iterations performed in the loop in App. A.2. There, we see fast convergence to an improved plan.

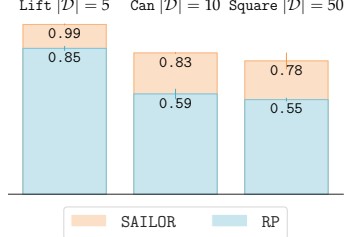

Figure 10: We find that explicit residual planning out-performs a Residual Policy (RP) across the board.

## 5  Discussion

In summary, we see that across a dozen visual manipulation tasks, SAILOR is able to train robust agents that recover from from the failures of the base policy via test-time planning and match the expert's outcomes, all without on-policy human feedback. Furthermore, scaling up the amount of data used to train a diffusion policy via behavioral cloning by $\approx$ 5-10$\times$ is unable to close this performance gap. We also see that our learned RM is able to detect nuanced failures, while the SAILOR agent is able to recover from them. As mentioned above, while for our experiments we use a base diffusion policy, MPPI planner, and Dreamer world model, the general SAILOR architecture is fully compatible with foundation behavior (e.g. VLAs) and world models trained on internet-scale data, which might unlock further levels of generalization and robustness – we leave this as a promising future direction.

Before we close, it is worth discussing two reasons why L2S is a particularly scalable algorithmic paradigm. The first is *data-efficiency*: as with any inverse RL approach that learns a *verifier* (i.e., reward model) from demonstrations rather than just a *generator* (i.e., policy), L2S requires less data than direct policy learning approaches on problems with a *generation-verification gap* [Swamy et al., 2025]. Given we have far less data in robotics than in domains like language modeling, squeezing as much as we can out of every demonstration is of paramount importance. Furthermore, as we scale up task horizons, the size of the generation-verification gap grows. This means that L2S techniques could exhibit better "data scaling laws" with task complexity than behavioral cloning techniques.

The second reason is *compute-efficiency*: in L2S, we only need to plan at the states *actually* encountered at test-time, rather than all states we could *potentially* encounter as in a train-time procedure. This provides provable computational benefits [Kearns et al., 2002]. Furthermore, expert iteration-style distillation lets us recycle this compute, so we never need to solve the same problem twice. Thus, L2S techniques could also exhibit better "compute scaling laws" than vanilla policy learning.

To close, we observe that in his oft-quoted essay *The Bitter Lesson*, Richard Sutton writes that:

> *"One thing that should be learned from the bitter lesson is the great power of general purpose methods ... The two methods that seem to scale arbitrarily in this way are **search** and **learning**."*

In SAILOR, rather than pick one, we fuse *learning* and *search* to chart a new course for imitation.

## Acknowledgments

We thank Kensuke Nakamura, Abigail DeFranco, and Andrea Bajcsy for their help in the early stages of this project and thoughtful advice throughout its completion. We thank Mohan Kumar for his help with setting up our demonstration collection interface and Yilin Wu for her advice on world models and demo collection. We thank Kensuke Nakamura, Harley Wiltzer, Jesse Farebrother, and Andrea Bajcsy for feedback on our initial draft of this paper. GKS is supported by a STTR grant. SC is supported in part by a Google Faculty Research Award, an OpenAI SuperAlignment Grant, an ONR Young Investigator Award, NSF RI #2312956, and NSF FRR #2327973. AJ is supported by Fonds de Recherche du Quebec (FRQ) (DOI assigned: https://doi.org/10.69777/350253), Calcul Quebec, and Canada Excellence Research Chairs (CERC) program.

## Contribution Statements

- **AJ** co-lead the implementation of all components, supervised the work of other students and helped write the paper. **VM** co-lead the implementation of all components and helped write the paper.
- **SK** implemented the first-pass version of our planner. **SK**, **JR**, and **YF** and implemented and ran experiments for our model-free inverse RL baseline. **JR** also helped write the paper.
- **AB** collected the demonstrations used for training all methods.
- **SC** helped advise the project, shaping its direction and presentation, suggested experiments and ablations, and provided computational and personnel resources.
- **GS** initially conceived of the project and algorithmic paradigm, gathered the team of students, managed the project over several phases, made all the figures, and wrote most of the paper.

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

# A  Additional Results

We now discuss two additional ablations of the `SAILOR` stack.

## A.1  Warm-Start Fraction

The warm-start fraction refers to the portion of total interaction budget we use to collect data with just the base policy (i.e., not performing any policy updates or using the planning stack). We then use this data to train the `WM`, `RM` and Critic. We performed an ablation of this fraction on the Square task with $|\mathcal{D}| = 50$ demonstrations.

In Fig. 11, we observe that using a warm-start fraction of 0.1 improves performance over not warm-starting. However, performance plateaus after using warm-start fraction of 0.2, which we used for all of our experiments.

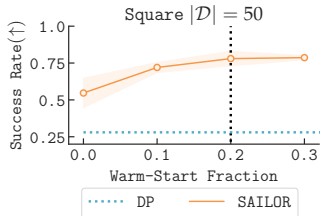

Figure 11: We ablate the warm-start percentage and find performance plateaus after 0.2. We use a total interaction budget of 500k environment steps and measure performance across three seeds.

## A.2  Residual Search vs. Residual Policy

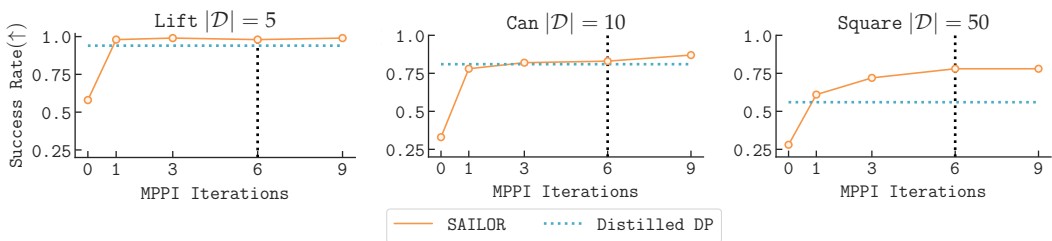

Figure 12: We use 6 iterations of the MPPI planning process (black line). We see that more iterations lead to higher performance up to 6 and do not see a degradation in performance even with 9 iterations. Iteration 0 represents the performance of the base `DP`. We also explored a post-hoc distillation step (blue line) for reducing wall-clock time. Doing so matches `SAILOR` on Lift and Can but not Square.

In Fig, 12, we compare the performance of `SAILOR` with different number of iterations of the MPPI planning process. We find that the performance converges after 2-3 iterations for most tasks and there is only a marginal improvement in later iterations, suggesting we could truncate `SAILOR` 's search process early. Doing so adaptively on a state-wise basis is an interesting future research direction.

Additionally, we measured that `SAILOR` spends  0.014s time per iteration of MPPI, while it takes 0.066s for the base diffusion policy to generate actions, making test-time planning a somewhat limited overhead. This is likely because the MPPI planner operates on top of lower-dimensional latent states from the world model, rather than high-dimensional image observations like the base `DP`.

Lastly, we explored performing a final post-hoc distillation of the `SAILOR` agent into the base policy (`Distilled DP`) for fast inference at test-time. As seen in Fig. 12, the performance of `Distilled DP` is similar to the `SAILOR` agent on `Lift` and `Can` tasks. However, on the harder `Square` task, test-time search outperforms the `Distilled DP`. This is potentially due to the limited capacity of the diffusion policy for representing the full planning circuit for sufficiently complex tasks.

# B  Implementation Details

In this section we first describe the network architecture for each component of SAILOR in Sec. B.1 following details of training pipeline and hyperparameters in Sec. B.2

## B.1  Network Architecture

**Base Policy.** SAILOR uses a Diffusion Policy (DP, Chi et al. [2023]) as the base policy. The base policy takes the stack of current and previous observations ($[o_{t-1}, o_t]$) as input and predicts the $k$-step action plan $a_{t:t+k}^{\mathsf{base}}$. The observation $o_t$ has proprioceptive states and RGB images from a wrist camera and a front camera. To encode RGB images, DP uses a ResNet-18 encoder initialized with the ImageNet1K_V1 weights, and the intermediate activations are passed through a Spatial Softmax layer [Mandlekar et al., 2021a] to get the final image embedding. Here, each input image uses a different copy of the encoder, and all encoded image inputs are concatenated together with the proprioceptive state and finally passed to the noise network $\epsilon$.

The noise network is a conditional 1D U-Net, where conditioning is incorporated via FiLM modulation, following the approach of [Chi et al., 2023]. The network is conditioned on the diffusion timestep (represented as a 16 dimentional embedding generated by a sinosuidal encoding layer followed by a small MLP), and the encoded observations. The UNet consists of multiple convolution and transposed convolution layers with channels [64, 128, 256], kernel size 3, GroupNorm Wu and He [2018] for stable training and Mish [Misra, 2019] activation function. The noise network is trained to reverse the forward noising process of adding Gaussian noise at each step (DDPM, Ho et al. [2020], Nichol and Dhariwal [2021]). Lastly, DP is trained using a behavior cloning objective, formulated as a denoising task where the model learns to predict the noise added to a $k$-step action chunk, conditioned on the given observation and diffusion timestep.

During inference, DP uses the DDIM sampling [Song et al., 2020] with the noise network to generate the action chunk for a given observation. In this work, we use the first action generated $a_t^{\mathsf{base}}$ to step in the environment. The agent uses a action blending mechanism that smoothens the action at current step using the predictions from earlier observations [Dasari et al., 2024]. We found this to significantly boost the performance of DP in our experiments.

**World Model.** The world model used in SAILOR uses the architecture of DreamerV3 [Hafner et al., 2024]. The encoder (enc) encodes the pixel-based inputs with stride 2 convolutions to a resolution of $4 \times 4$ and state inputs are embedded with a 5-layer MLP. Note that the WM is using a separate encoder for observations as we found it to work well in our experiments. The decoder (dec) uses a transposed convolutions with stride 2 to reconstruct image observations and a 5-layer MLP to reconstruct the proprioceptive states. Note that the RGB images are downscaled to size $64 \times 64$ for training. The latent representation $z_t$ is a combination of a deterministic recurrent state $h_t$ and a stochastic state $s_t$. The deterministic state is has a GRU network with 512 dimensional hidden state. The latent state $z_{t-1}$ and action $a_{t-1}$ from previous time step are used to estimate the deterministic component $h_t$ at current step. The deterministic state is fed through the dynamics model f to sample the prior stochastic state $\hat{z}_t$. Moreover, the deterministic state combined with the observation is passed through the encoder enc to get the posterior state $z_t$. The stochastic representation of 1024 dimensions is sampled through a vector of multinomial distributions where the gradients are backpropagated through straight-through estimators [Bengio et al., 2013] while learning.

The world model is trained by hybrid learning where the half of the batch of sequences is obtained from the demonstration $\mathcal{D}$ and other half comes from the replay buffer $\mathcal{B}$. More formally, consider a sampled batch of observation subsequences $o_{t:t+u}$, actions $a_{t:t+u}$, continuation flag $c_{t:t+u}$ (to predict the end of episode), where $t \sim \mathcal{U}\{1, \ldots, T\}$, $u$ denotes the length of the sequence, and $T$ denotes the length of a trajectory. We minimize a combination of a prediction loss $\ell_{\mathsf{pred}}$, a dynamics loss $\ell_{\mathsf{dyn}}$ and a representation loss $\ell_{\mathsf{rep}}$ over samples drawn from $\mathcal{D}$ and $\mathcal{B}$:

$$\ell(\mathcal{D}, \mathcal{B}, \mathtt{WM}) = \frac{1}{2}\mathbb{E}_{o_{t:t+u}, a_{t:t+u}, c_{t:t+u} \sim \mathcal{D}}[\beta_{\mathsf{pred}}\ell_{\mathsf{pred}}(\cdot) + \beta_{\mathsf{dyn}}\ell_{\mathsf{dyn}}(\cdot) + \beta_{\mathsf{rep}}\ell_{\mathsf{rep}}(\cdot)]$$

$$+ \frac{1}{2}\mathbb{E}_{o_{t:t+h}, a_{t:t+h}, c_{t:t+h} \sim \mathcal{B}}[\beta_{\mathsf{pred}}\ell_{\mathsf{pred}}(\cdot) + \beta_{\mathsf{dyn}}\ell_{\mathsf{dyn}}(\cdot) + \beta_{\mathsf{rep}}\ell_{\mathsf{rep}}(\cdot)], \quad (3)$$

where $\beta_{\mathsf{pred}} = 1.0$, $\beta_{\mathsf{dyn}} = .1$ and $\beta_{\mathsf{rep}} = .5$ represent the loss weights. The loss functions are:

$$\ell_{\mathsf{pred}}(o_{t:t+u}, a_{t:t+u}, c_{t:t+u}, \mathtt{WM}) = -\sum_{h=t}^{t+u} \ln \mathtt{dec}(o_h|z_h) - \ln \mathtt{dec}(c_h|z_h),$$

$$\ell_{\mathsf{dyn}}(o_{t:t+u}, a_{t:t+u}, c_{t:t+u}, \mathtt{WM}) = \sum_{h=t}^{t+u} \max(1, \mathbb{D}_{\mathrm{KL}}[\mathtt{sg}(\mathtt{enc}(z_h|z_{h-1}, a_{h-1}, o_h))\|\mathtt{dec}(\hat{z}_h|z_{h-1}, a_{h-1})]),$$

$$\ell_{\mathsf{rep}}(o_{t:t+u}, a_{t:t+u}, c_{t:t+u}, \mathtt{WM}) = \sum_{h=t}^{t+u} \max(1, \mathbb{D}_{\mathrm{KL}}[\mathtt{enc}(z_h|z_{h-1}, a_{h-1}, o_h)\|\mathtt{sg}(\mathtt{dec}(\hat{z}_h|z_{h-1}, a_{h-1}))]).$$

The prediction loss optimizes for reconstructing the observations via a Mean Squared Loss (MSE) criterion and the continuation predictor via logistic regression. The dynamics and representation losses optimize the same objective is optimized with different set of parameters: the former updates the dynamics model to predict the posterior state while the latter term ensures that the stochastic term is more predictable. Note that the dynamics and representation loss only differ in the stop-gradient term $\mathtt{sg}$ and the loss weight terms.

**Reward Model.** The reward model $\mathtt{RM}$ takes the latent representation $z_t$ as input and uses a 2-layer MLP to predict a scalar value expressing the desirability of being in a state. The $\mathtt{RM}$ is optimized with the loss function defined in Eq. 1. The $\mathtt{RM}$ is updated with a gradient penalty term [Gulrajani et al., 2017] with a coefficient of 10. To stabilize learning, the $\mathtt{RM}$ is updated less frequently than the world model. Moreover, we do not update the parameters of the world model with the loss of the $\mathtt{RM}$.

**Critic Network.** Similar to $\mathtt{RM}$, the critic $\mathtt{V}$ network uses a 2-layer MLP and predicts the discounted average rewards of future states with the latent representation $z_t$ as input. The critic is updated with the Mean Squared Loss (MSE) with target $\mathbf{v}_t^\lambda$ defined in Eq. 2:

$$\ell(z_{t:t+u}, \mathtt{V}) = \sum_{h=t}^{t+u} (\mathtt{V}(z_h) - \mathbf{v}_h^\lambda)^2. \tag{4}$$

Unlike $\mathtt{RM}$, the critic is updated with the $\mathtt{WM}$. Training the critic more frequently than $\mathtt{RM}$ is important to ensure it accurately estimates the average rewards and remains synchronized with the changing reward function. Similar to Hansen et al. [2024], $\mathtt{SAILOR}$ maintains and uses an ensemble of 5 value networks. Like Dreamer, $\mathtt{SAILOR}$ also maintains a slow value network to compute a slow target for the critic network and uses this as a regularizer for critic loss. The slow networks are updated with EMA over the critic parameters.

**MPPI Planner.** $\mathtt{SAILOR}$ uses MPPI for planning withing the world model in this work. The planner maintains a gaussian distribution with mean $\mu$ and diagonal covariance $\sigma$ to predict the residual action $\Delta_{t:t+k}^*$. The planning procedure is described in Alg. 1 where the parameters are initialized with 0 mean and a fixed standard deviation. At each iteration, 256 action chunks are sampled and scored with the $\mathtt{RM}$ and $\mathtt{V}$ by imagining future latent with $\mathtt{WM}$. The top 64 sequences with highest n-step returns is used to update the parameters $\mu$ and $\sigma$. Unrolling in the latent space allows evaluating large batches in parallel on a single GPU, and thereby makes planning efficient. After 6 iterations, an action plan $\Delta_{t:t+k}^\star$ is sampled using final parameters $\mu^\star$ and $\sigma^\star$.

### B.2 Training details

With the demonstrations $\mathcal{D}$, $\mathtt{SAILOR}$ first pretrains the base policy– DP. The training procedure of $\mathtt{SAILOR}$ was outlined in Alg. 2. The pretrained DP is used to collect rollouts in the environment and uses around $20\%$ of total environment steps. For instance, when the agent is tasked with 100K environment steps, the pretrained DP is deployed to collect for 20K transitions. $\mathtt{SAILOR}$ adds a small noise sampled from $\mathcal{N}(0, .1)$ to the action to promote exploration while collecting data from the environment. The agent maintains a uniform replay buffer $\mathcal{B}$ with an online queue of size $1 \times 10^5$. During the *warmstart phase*, the $\mathtt{WM}$, $\mathtt{RM}$ and critic $\mathtt{V}$ are updated with a hybrid batch sampled from demonstration $\mathcal{D}$ and replay buffer $\mathcal{B}$. Note that, the gradient steps of $\mathtt{WM}$ and $\mathtt{V}$ was set to $1.5\times$ the number of transitions collected. For training stability, the $\mathtt{RM}$ is updated slowly and once every 100 updates to the $\mathtt{WM}$. After warmstart, $\mathtt{SAILOR}$ uses online finetuned with batch data collection and updates over multiple rounds. At each round, $\mathtt{SAILOR}$ deploys the planner to collect on-policy

trajectories for 3500 environment steps. The `WM` and `V` are then updated for 5000 gradient steps where the `RM` is updated once every 100 gradient steps of `WM`. After every 10 rounds, the last 64 collect trajectories are relabeled with the planner and the base policy is updated for 1000 iterations using a hybrid batch composed of relabeled data and expert demonstrations . In Table. 1, we provide the details of hyperparameters used in this work. Our agents where trained on a single NVIDIA 6000Ada GPU with 48 GB memory and takes 36 hours for training end-to-end with 500K environment steps.

## C    Baselines

For baselines, we compare `SAILOR` with the pretrained DP policy and a diffusion based IRL method (`DPPO-IRL`). Specifically, we apply DPPO [Ren et al., 2024a] – a model-free RL algorithm that directly updates DP parameters – with feedback from a reward model learned in the same as for `SAILOR` (Eq. 1). In contrast to learning a `WM` and learning to search in `SAILOR`, `DPPO-IRL` optimizes the policy directly using the outcomes of the learned reward model. To encode the observation, the reward model used the same encoding as the base DP of DPPO and a 2-layer MLP of width 256. We tuned the hyperparameters of the reward model (update epochs and batch size), and observed that the best version used 2 as update epochs and a batch size of 100. As for `SAILOR`, we used a gradient penalty term with a coefficient of 10 to stabilize learning of the reward model.

| Name | Value |
|---|---|
| **DP Pretraining** | |
| Batch Size | 256 |
| Optimizer | AdamW |
| Training iterations | 24,000 |
| LR scheduler | Cosine Annealing |
| LR scheduler warmup steps | 100 |
| LR range | $[1 \times 10^{-4}, 1 \times 10^{-5}]$ |
| **World Model** | |
| Replay capacity | $1 \times 10^5$ |
| Batch size | 16 |
| Batch length | 32 |
| Optimizer | Adam |
| Reconstruction Loss Scale | 1.0 |
| LR | $1 \times 10^{-4}$ |
| Demo Sampling Ratio | 50% |
| **Reward Model** | |
| Optimizer | Adam |
| LR | $3 \times 10^{-5}$ |
| Gradient Penalty coefficient | 10 |
| **Critic** | |
| Discount factor $\gamma$ | .997 |
| Return lambda $\lambda$ | .95 |
| EMA regularizer | 1 |
| EMA decay | .98 |
| Optimizer | Adam |
| LR | $3 \times 10^{-5}$ |
| Ensemble size | 5 |
| **MPPI Planner** | |
| Iterations | 6 |
| Samples | 256 |
| Top candidates | 32 |
| Temperature | .5 |
| **General** | |
| Warmstart env step ratio | 20% |
| Env steps per round | 3500 |
| Update steps per round | 5000 |
| Distillation frequency | 10 |
| Trajectories relabeled for distillation | 64 |
| Distillation steps | 1000 |

Table 1: **Hyperparameters.** For training, we recommend tuning training parameters in the General section that includes the update steps per round $\in [1000, 2000, 3500, 5000, 10000]$, distillation frequency of the base policy $\in [1, 5, 10, 50]$ and distillation steps of the base policy $\in [500, 1000, 2000]$. LR, Env and EMA denotes learning rate, environment and exponential moving average.

# D  Benchmarks

In this section, we describe the environments used in this work. The experiments are conducted on multiple robotic manipulation environments from RoboMimic [Mandlekar et al., 2021b], Robo-Suite [Zhu et al., 2020] and ManiSkill3 [Tao et al., 2025]. Fig. 13 presents a visual image for each of the task used in this work. For each task, the agent is provided with an RGB image from the wrist camera and an RGB image from a camera in front of the agent. The agent is also given the proprioceptive states composed of position, orientation of the end effector and the position of gripper. In Table 2, we provide the episode horizon, the environment steps used for IRL training and a brief description of the task. The action space is a 7-dimensional vector with values between $[-1, 1]$. The first 6 dimensions of the action control the change in position and orientation of the end-effector and the last value opens or closes the gripper.

| Domain | Task | Horizon | Env Steps | Description |
|---|---|---|---|---|
| RoboMimic | **Lift** | 100 | $1 \times 10^5$ | Lift Block above the desk. |
| | **Can** | 200 | $5 \times 10^5$ | Lift can and place in correct bin. |
| | **Square** | 200 | $5 \times 10^5$ | Pick square tool and insert in slot. |
| RoboMimic | **Door** | 200 | $1 \times 10^5$ | Pull down handle and open door. |
| | **Stack** | 150 | $3 \times 10^5$ | Lift block and place above other block. |
| | **Bread** | 200 | $3 \times 10^5$ | Lift bread and place in correct bin. |
| | **Cereal** | 150 | $5 \times 10^5$ | Lift cereal and place in correct bin. |
| | **Round** | 300 | $5 \times 10^5$ | Pick round tool and place in slot. |
| | **BreadCan** | 400 | $5 \times 10^5$ | Place both objects in respective bins. |
| ManiSkill | **PullCube** | 50 | $1 \times 10^5$ | Pull cube to the marked area |
| | **PokeCube** | 100 | $3 \times 10^5$ | Use tool to poke cube to marked area |
| | **LiftPeg** | 150 | $3 \times 10^5$ | Lift peg to make it stand upright |

Table 2: The maximum episode length (Horizon), environment steps and description of the tasks.

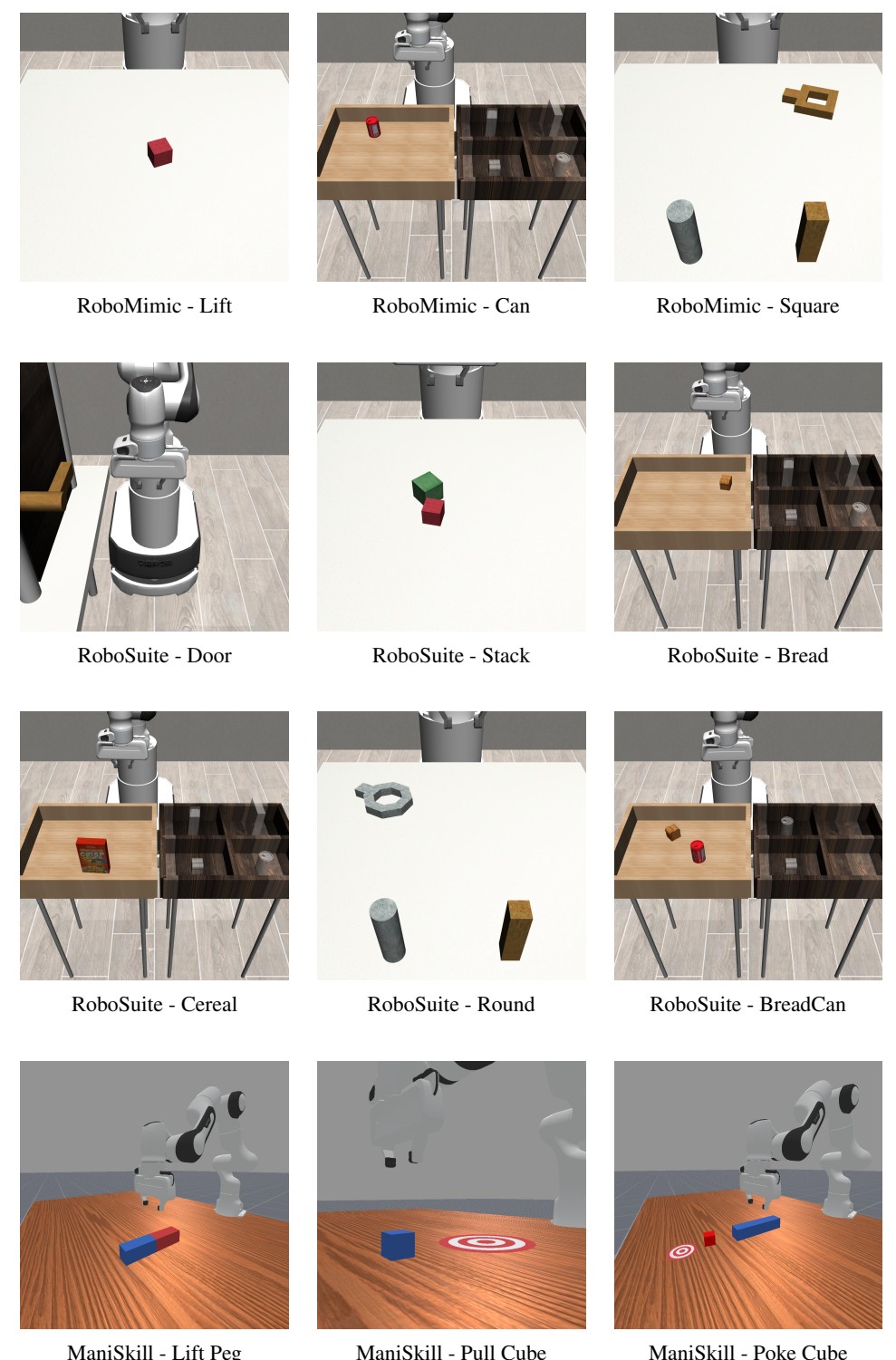

Figure 13: Visual description of all tasks used from RoboMimic (top row), RoboSuite (middle rows) and ManiSkill (bottom row).

# E  Demonstrations

In this section, we describe the demonstrations used for learning. For RoboMimic tasks, we used the demonstrations provided in the dataset. For tasks in RoboSuite and ManiSKill, we collected upto 200 demonstrations using a 3D SpaceMouse. The camera angles were adjusted for each task to make the process intuitive to the human teleoperator. Below we provide a table of the number of demonstrations used per task for Fig. 4. For BreadCan, Cereal and Round we collected upto 200 demonstration for our result in Fig. 5. We plan to release the demonstrations and the codebase.

| Domain | Task | Number of Demonstrations |
|--------|------|--------------------------|
| RoboMimic | Lift | 5, 10, 15 |
| | Can | 10, 15, 20 |
| | Square | 50, 75, 100 |
| RoboMimic | Door | 1, 3, 5 |
| | Stack | 15, 20, 25 |
| | Bread | 15, 20, 25 |
| | Cereal | 20, 25, 30 |
| | Round | 25, 50, 75 |
| | BreadCan | 50, 75, 100 |
| ManiSkill | PullCube | 10, 15, 20 |
| | PokeCube | 5, 10, 15 |
| | LiftPeg | 15, 20, 25 |

Table 3: The number of demonstrations used for each task.

# F  Related Work (Extended)

**Reward Models.** Another approach to interactive imitation learning is *inverse reinforcement learning* (IRL, [Ng et al., 2000, Syed and Schapire, 2007, Ziebart et al., 2008, Ho and Ermon, 2016, Swamy et al., 2021]), which does not require human-in-the-loop queries. In IRL, one learns a classifier that maximally differentiates learner from expert behavior and uses it as a *reward model* (RM) for RL-based policy updates. Different forms of reward models include successor features [Jain et al., 2025b], score matching [Wu et al., 2025a], and optimal transport metrics Haldar et al. [2023b,a]. Unfortunately, the RL step of IRL is often rather interaction-inefficient. Recent theoretical work has argued that rather than a *global* RL procedure, a *local search* procedure[6] is sufficient for performant imitation [Swamy et al., 2023, Espinosa-Dice et al., 2025]. SAILOR fits within the overarching algorithmic paradigm of local search IRL but *learns to search* rather than learning a policy directly.

**World Models.** World models (WMs, Ha and Schmidhuber [2018]) have been an integral component of impressive RL results in a variety of domains [Hafner et al., 2020, 2021, Jain et al., 2022, Hansen et al., 2022, Hafner et al., 2024, Hansen et al., 2024, Zhou et al., 2025, Bruce et al., 2024, Parker-Holder et al., 2024, Agarwal et al., 2025], including real-world robotics [Mendonca et al., 2021, Wu et al., 2023]. While our overall algorithmic framework is agnostic to the choice of WM architecture, we use Dreamer-style world models [Hafner et al., 2020] in our experiments due to their ubiquity. Popov et al. [2024] use rollouts in a learned world model to minimize a trajectory-level divergence between the expert and the learner at train time on autonomous driving problems. In contrast, we use the world model at test-time to enable recovery from just the mistakes the learner actually makes, which might be more computationally efficient than a global search procedure [Kearns et al., 2002].

---

[6]By *local search*, we mean merely competing with the expert rather than the optimal policy for an adversarially chosen reward. For the former, there exist algorithms that avoid the worst-case, exponential-in-the-horizon exploration complexity of RL [Bagnell et al., 2003, Ross and Bagnell, 2014, Swamy et al., 2023].

