# OpenReview forum: "A Smooth Sea Never Made a Skilled SAILOR: Robust Imitation via Learning to Search"
_NeurIPS.cc/2025/Conference — NeurIPS 2025 spotlight_

### Official Review · Reviewer_kASh · 2025-06-26

**Clarity:** 3
**Significance:** 3
**Originality:** 3
**Rating:** 5
**Confidence:** 3

**Summary:**

The paper proposes a novel architecture and training scheme for learning robust policies from expert demonstrations. In particular, the paper learns a latent-space (“world”) model to simulate rollouts, a reward function to quantify the quality of latent states, and a critic to estimate the residual value of truncated latent-state trajectories. These three components can be used to plan the next action (via MPPI in the paper). Additionally the paper uses a base policy (Diffusion based) for more efficient search, and to distill spent test-time compute spent on searching good actions back into the base policy. The paper demonstrates that this leads to good performance across a range of simulated robotic manipulation tasks with relatively few expert demonstrations. The method performs significantly better than naive behavioral cloning, even if the available number of demonstrations for the latter is increased by a factor of 5 to 10. The main premise is that the method addresses one of the main failure cases of behavioral cloning, which is going off distribution, by having the ability to recover by planning via the learned components (“learning to search”).

**Questions:**

I have no major concerns or questions (please see my comments on “main weaknesses” above, which I do not consider to be major). Here are two very minor comments:

1. The second fundamental problem with expert imitation on partially observable decision-making tasks (besides having no off-distribution coverage) is that models need to learn to update their belief state differently when it is the result of their own action, compared to observing an expert action. Technically, the model’s own actions must be treated as interventions, whereas expert actions can be treated via standard probabilistic conditioning. In BC, models cannot learn this (as all actions are conditioning), and are thus likely to suffer from self-delusions (see: Shaking the foundations: delusions in sequence models for interaction and control, Ortega et al. 2021). When actively interacting (whether model based or not) models naturally learn the correct update mechanisms. The proposed method in the paper thus also seems to address this second, and quite subtle, fundamental problem with BC in partially observed interactive environments (where experts may hold unobserved “privileged” belief states).
2. Though the paper does not introduce the term “Learning to Search”, I think it is not ideal, as it suggests imitating an expert policy where search emerges purely from learning (e.g., BC). Rather, the method in the paper is explicitly programmed for “Searching with Learned Models”. No strict need to change this in the paper, but maybe it’s still time to stop the proliferation of a misleading term.

**Ethical Concerns:**

["NO or VERY MINOR ethics concerns only"]

**Final Justification:**

I had no major issues and only minor clarifying questions/comments after the initial round of reviews, and the authors have nicely addressed all of them. I therefore continue to argue in favor of accepting the paper and have raised my confidence (still relatively low as I am no expert in robotics and do not precisely know the SOTA methodologies).

**Limitations:**

The paper checklist for the discussion of limitations is answered with “NA” (which corresponds to “The answer NA means that the paper has no limitation”). I don’t think that this is what the authors meant, and I would encourage them to add a brief discussion of limitations, particularly w.r.t. the generality of the current findings and scaling up the method. Alternatively, please add a justification for the answer to the paper checklist.

**Paper Formatting Concerns:**

No concerns.

**Quality:**

3

**Strengths And Weaknesses:**

**Main contributions**
1. A non-trivial combination of previously proposed and used parts, techniques, losses, etc. The choice of many of these components is discussed and contrasted against alternatives, and making all these components play well together is far from being guaranteed.
2. Good empirical results, particularly against naive BC (even at increased amounts of data) and a version of inverse RL (DPPO).
3. Ablations for the main parts of the method, and some behavioral analysis of the ability to recover from off-distribution situations.
4. The paper is generally well written and easy to follow (though, given the number of parts of the model certain sections must be dense to stay within the page limit).

**Main weaknesses**
1. Part of the writing (particularly early in the paper) sounds as if the combination of a learned world model, reward model, and a planner (with base policy) had never been used before. This is clarified in the related work section, which makes it more clear that this particular combination of these parts is novel (and I agree it is non-trivial), along with a number of losses / training schemes for parts to make everything work together. Since I am not an expert in robotics, I cannot confidently judge the novelty and originality (and have lowered my confidence accordingly), but I would be surprised if there are no comparable roughly similar approaches in the literature.
2. Related to 1. I am wondering whether there are stronger baselines that one could compare against, particularly methods that have a similar aim (planning via learned model / reward function). I may be wrong, and leave it to the expert reviewers to point out any such methods. I do appreciate the naive BC baseline to support the main claims and the inverse RL baseline is also a good addition.
3. The paper is dense and is missing a discussion and conclusion and a limitations section. While I understand that the page limitations require a trade-off, and am happy that the many parts and pieces are explained well and important ablations are shown in the main paper, not having these additional sections is a small weakness (I do not expect a change to the main paper; I am merely pointing out a small weak point).

**Verdict:**

Based on the empirical results, the non-trivial combination of a number of components and the protocol to make these components play well together, the ablations, and the good writing, the paper looks solid and ready for publication to me. As stated above, I am not an expert in robotics, so I cannot confidently judge the novelty, originality, and potential impact. I am therefore currently between a weak accept and an accept, leaning towards the latter. I am particularly uncertain whether the method can relatively simply be put together from the current literature and whether strong baselines are missing, but hopefully other reviewers will address this. Currently, I have no reasons to recommend rejecting the paper.

---

> ### Author Rebuttal · Authors · 2025-07-31
>
> Thank you for your thoughtful review of our work. We are glad that you found the integration of multiple learned components compelling and appreciated the thoroughness of our experiments. We address your questions below:
>
> > W1: Since I am not an expert in robotics, I cannot confidently judge the novelty and originality (and have lowered my confidence accordingly), but I would be surprised if there are no comparable, roughly similar approaches in the literature.
>
> We thank the reviewer for their honesty here. We agree that while subsets of using a learned latent-space world model, reward model, planner, and base policy have been explored in prior work (as referenced in our related works section), we struggled to find work in which *all* of these components were used together. Furthermore, in much of the prior work, some of these components are learned via auxiliary sources of information (e.g., ground-truth reward labels) or need access to lower-dimensional state representations. In contrast, **we believe one of the key novelties of our framework is that all of these components can be learned just from visual demonstrations / on-policy rollouts, with no reward labels / further human supervision required**.
>
> > W2: Related to 1. I am wondering whether there are stronger baselines that one could compare against, particularly methods that have a similar aim (planning via learned model/reward function). I may be wrong, and leave it to the expert reviewers to point out any such methods. I do appreciate the naive BC baseline to support the main claims, and the inverse RL baseline is also a good addition.
>
> We agree that more baselines could only make our work stronger. As suggested by reviewer AkFH, we added in a baseline where, rather than learning a residual planner, we instead learned a residual *policy*, allowing us to ablate the benefit that explicit planning at test time provides. We trained this policy in standard DREAMER fashion (i.e., via backprop through the world model). Our experiments in the table below (averaged with 3 seeds per task) show that **SAILOR with test-time planning is better than learning a residual actor**. We have added these results to our paper.
>
> Second, we'd like to make a note on the strength of our inverse RL baseline. The reason we used DPPO as the policy optimizer for our inverse RL baseline is that it has been shown to significantly outperform other RL algorithms for training diffusion policies (e.g., DQL, IDQL, QSM) and demonstrated strong performance with ground-truth rewards on the Robomimic tasks we consider [1].
>
> | | Lift | Can | Square |
> |----------|----------|----------|----------|
> | DP |.58    | .33    | .28    |
> | WM + RM + Residual Policy|.85    | .61    | .54    |
> | SAILOR|.99    | .82    | .78    |
>
> > W3: The paper is dense and is missing a discussion and conclusion, and a limitations section.
>
> We agree with the reviewer that adding an explicit discussion and limitation section would further strengthen our manuscript. We are happy to move this to the additional page for camera-ready or to the appendix. In particular, we have added the following:
>
> ### Limitations
> At a high level, we believe there are two key limitations in our work that we believe offer promising opportunities for future work. The first is to perform experiments on real robots -- recent work FOREWARN [2] has shown that similar ideas to SAILOR can be scaled to real robots via the combination of diffusion policies, the Dreamer WM architecture, using VLMs as reward functions, and test-time planning. We believe that adding in other components of the SAILOR architecture (e.g., the use of a residual search rather than mode selection as in FOREWARN) could further improve performance. Second, we currently train the world model from scratch. Instead, leveraging a pre-trained foundation world model like Dino WM or V-JEPA could further improve sample efficiency and potentially eliminate the need for a warm start.
>
> ### Discussions
> In summary, we see that across a dozen visual manipulation tasks, SAILOR is able to train robust agents that recover from the failures of the base policy via test-time planning and match the expert’s outcomes, all without any additional human data. Furthermore, scaling up the amount of data used to train a diffusion policy via behavior cloning by ≈ 5-10× is unable to close this performance gap. We also see that our learned RM is able to detect nuanced failures, while the SAILOR agent is able to recover from them. As mentioned above, while for our experiments we use a base diffusion policy, MPPI planner, and Dreamer world model, the general SAILOR architecture is fully compatible with foundation behavior (e.g., VLAs) and world models trained on internet-scale data, which might unlock further levels of generalization and robustness – we leave this as a promising future direction.
>
> > Q1:  The second fundamental problem with expert imitation on partially observable decision-making tasks (besides having no off-distribution coverage) is that models need to learn to update their belief state differently when it is the result of their own action, compared to observing an expert action. Technically, the model’s own actions must be treated as interventions, whereas expert actions can be treated via standard probabilistic conditioning. In BC, models cannot learn this (as all actions are conditioning), and are thus likely to suffer from self-delusions (see: Shaking the foundations: delusions in sequence models for interaction and control, Ortega et al. 2021). When actively interacting (whether model based or not) models naturally learn the correct update mechanisms. The proposed method in the paper thus also seems to address this second, and quite subtle, fundamental problem with BC in partially observed interactive environments (where experts may hold unobserved “privileged” belief states).
>
> Yes -- in fact, it *provably* does under some assumptions (see [3] for more)! In greater detail, **interactive approaches to imitation -- both DAgger and inverse RL -- are able to handle unobserved confounders (e.g., context variables that only the expert can see) as the learner actually *intervenes* in the environment by taking actions and observing their consequences**, rather than learning from off-policy / "observational" data. In contrast, at test time, a BC policy treats the learner's past actions as though they were from the *expert*, which leads to the "self-delusions" / compounding errors the reviewer mentions. We mentioned partial observability as one possible cause of compounding errors in our related works and have added Pedro's lovely paper to the corresponding section.
>
> > Q2: Though the paper does not introduce the term “Learning to Search”, I think it is not ideal, as it suggests imitating an expert policy where search emerges purely from learning (e.g., BC). Rather, the method in the paper is explicitly programmed for “Searching with Learned Models”. No strict need to change this in the paper, but maybe it’s still time to stop the proliferation of a misleading term.
>
> We thank the reviewer for their comment. In some sense, we believe that the term "learning to search" is not entirely inaccurate, in the sense that we're learning the components required to execute a search algorithm at test-time. Furthermore, we believe that an interesting direction for future work could be to replace the fixed MPPI algorithm with a learned operator to combine the learned world and reward models, and that SAILOR serves as an important first step towards doing so. That said, we do agree that we could add more clarification about what we mean by the term in our work (i.e., using learned models as part of a fixed search algorithm) and would be happy to do so, say, in the introduction.
>
> > Limitations
>
> As described above, we have added discussion and limitations sections, as well as updated the paper checklist accordingly.
>
> We hope our responses and proposed changes have resonated with you and hope you consider increasing your score.
>
> Best Regards,\
> The Authors
>
> #### References
> [1] Ren et al., Diffusion Policy Policy Optimization, ICLR 2025.\
> [2] Wu et al., From Foresight to Forethought: VLM-In-the-Loop Policy Steering via Latent Alignment, RSS 2025.\
> [3] Swamy et al., Sequence Model Imitation Learning with Unobserved Contexts, NeurIPS 2022.

---

> > ### Comment · Reviewer_kASh · 2025-08-04
> > **Thanks for the detailed comments and clarifications**
> >
> > Since I had no major issues and only minor questions I did not consider any of these points necessary for recommending acceptance. Nonetheless, the authors have nicely clarified my questions or low-confidence statements, and I appreciate the addition of a brief limitations and discussions section (which, I think, would go well on the extra page of the camera ready). Taking all information together, including the other reviews and responses, my rating remains the same (clear accept) and I have increased my confidence.

---

### Official Review · Reviewer_FQJB · 2025-07-01

**Clarity:** 4
**Significance:** 2
**Originality:** 3
**Rating:** 4
**Confidence:** 3

**Summary:**

The paper proposes SAILOR, an imitation learning approach that aims at overcoming the usual out-of-distribution errors of behavior cloning by "learning to search" on top of learned world and reward models. The idea is to train these models on a mixture of expert data and data generated from a base policy (trained via BC). At inference time, SAILOR performs residual planning via MPPI to refine the plans proposed by the base policy, hence enabling eg recovering from failures. The method is tested on top of diffusion policies (DPs) and shown to perform better than plain BC (on top of DPs) and then a model-free IRL method (DPPO-IRL) on 12 complex visual manipulation tasks.

**Questions:**

Please address limitation above

**Ethical Concerns:**

["NO or VERY MINOR ethics concerns only"]

**Final Justification:**

Reasons why I have increased my score from 3 to 4:
- The authors provided additional experiments that make the main conclusions convincing
- The authors also provided experiments to assess the scalability of the approach in terms of planning times

Reasons not to increase further:
- Testing only on top of diffusion policies is not sufficient to assess the generality and broad applicability of the proposed method
- I believe the fact that SAILOR has online access to the environment is the key factor to enable eg error recovery and makes the contribution of the main algorithmic components (eg WM + planning) milder. For instance, I don't expect those components to work that well if trained entirely on expert data (no online samples)

**Limitations:**

yes

**Quality:**

3

**Strengths And Weaknesses:**

Strengths:
1. The paper considers a relevant and well-motivated problem: BC algorithms are ubiquitous in practice (especially now in the era of LLMs/VLMs/VLAs) and their out-of-distribution issues are well known for sequential decision-making problems
2. The proposed algorithm is simple yet quite powerful. I particularly like the idea of including world modeling and planning to recover from mistakes and improve performance
3. The paper is extremely well written and easy to read. Pseudo-codes are clear and self-contained, while the figures do a great job in supporting the presentation
4. Existing literature is extensively discussed
5. Experiments are done on sufficiently complex tasks (12 visual manipulation problems from different suites). I particularly appreciate that the models take as input both proprioceptive and visual observations, which reflects real-world usecases

Weaknesses:
1. My main concern is about the experimental evaluation, as I found the results not fully convincing to support the main claims. In particular:
   -  "Can SAILOR outperform DP trained on the same D?" Unless I am missing something, SAILOR and DP are not trained on the same data: both use the expert demonstrations D, but SAILOR is additionally generating online rollouts in the environment. If this is the case, this makes the comparison pretty unfair, and especially makes it hard to conclude whether the ability of SAILOR to recover from failures really stems from world modeling and planning (as claimed) or whether it is simply because those situations are generated by rollouts at train time, so SAILOR has evidence to recover that DP does not have
   - "Does DP catch up with more data?" May have the same issue as above, as SAILOR always uses more data
   - "How much real-world interaction does the WM save us?" Given the many differences between SAILOR and DPPO-IRL, it is hard to conclude if this is due to the WM or other algorithmic choices. For instance, DPPO-IRL uses PPO as the base optimizer, which is known to be very sample inefficient, while SAILOR is closer to an off-policy method as it reuses efficiently previous data. I wonder if other baselines can shed more light on this (e.g., can an off-policy diffusion-based like Diffusion Q-learning be adapted as an IRL variant?)
2. The claim "Our key insight is that we can infer the latent search process required to recover from local mistakes from the same source of human data required for the standard behavioral cloning pipeline" in the introduction also seems to contradict a bit the functionality of the algorithm: as SAILOR can rollout data in the environment, how can we conclude that it can recover from mistakes from expert data only?
3. Section 3.1 mentions that "Beyond these benefits, the unique advantage of the L2S paradigm is a computational one: the ability to only need to plan at the states encountered at test-time". I was actually thinking the opposite, as planning methods are known to be computationally much more inefficient at test-time than pre-trained policies (of course if you ignore pre-training time, but that should be eventually much smaller than the cost of doing multiple test-time evaluations). I think one advantage is instead that planning implicitly expands the space of possible behaviors that can be generated (as opposed to a pre-trained policy which is limited to the capacity of a fixed neural net)
4. The paper mentions data SAILOR can be applied on top of any base policy, but then the method is tested only on top of DPs. To really show this generality, I'd be better to add experiments on other base policies (eg recent VLAs as mentioned in the text)

Overall I think this is a very promising paper which I really enjoyed reading, though I have to vote for rejection given the limitations above. I am more than happy to increase my score if I am missing something or if additional experiments are provided to reinforce the main conclusions.

---

> ### Author Rebuttal · Authors · 2025-07-31
>
> Thank you for your feedback, which helped strengthen our work. We are glad you enjoyed the simplicity of our approach and the ability of SAILOR to achieve good performance on a complex set of tasks having visual and proprioceptive inputs, a setting that reflects real-world use cases. We respond to your questions below.
>
> > W1 (a): "Can SAILOR outperform DP trained on the same D?" Unless I am missing something, SAILOR and DP are not trained on the same data: both use the expert demonstrations D, but SAILOR is additionally generating online rollouts in the environment. If this is the case, this makes the comparison pretty unfair, and especially makes it hard to conclude whether the ability of SAILOR to recover from failures really stems from world modeling and planning (as claimed) or whether it is simply because those situations are generated by rollouts at train time, so SAILOR has evidence to recover that DP does not have
>
> > W1 (b): "Does DP catch up with more data?" May have the same issue as above, as SAILOR always uses more data
>
> We believe that these two questions essentially boil down to asking whether it is fair to compare SAILOR to DP, given that the former gets access to online interaction while the latter does not. There are two points worth mentioning here. The first is that all methods (interactive and not) get access to the same set of expert demonstrations – it is standard practice in imitation learning to compare methods based on how they perform based on the same set of expert demonstrations, even when some are interactive (e.g., in the GAIL paper [1]). More generally, perhaps, the entire field of inverse RL (which includes SAILOR) focuses on interactively learning better policies from the same set of expert demonstrations that an offline algorithm like behavioral cloning uses. The second is that, given we do not assume access to any ground-truth reward information, online data provides no intrinsic information about how to recover. More explicitly, if we simply see a rollout and have no way to judge how good it is (i.e., whether it “recovers” or whether it even made a mistake in the first place), it is not clear how to use it to improve the performance of the policy. For example, if we know that we dropped an object but not that dropping an object is bad, it is unclear what to do with this information. It is only because we infer a reward model from demonstrations in SAILOR (and, more generally, in inverse RL) that we are able to use this on-policy data to learn how to recover. In short, (a) the comparison we make in this paper is extremely standard in the literature, and (b) merely providing an algorithm with access to online samples provides no intrinsic value.
>
> That said, there is a related question we find to be quite interesting: **How much does planning corrective actions against a learned RM + WM improve performance over using a residual policy that does not perform a test-time search** (and therefore may only handle situations seen at train-time). Both of these options get access to interaction, but only the former is able to respond to new situations unseen at training time. To compare these two options, we added an additional baseline in which we replaced the residual search process in SAILOR with a *residual policy*. More explicitly, the residual actor network is tasked to predict a corrective action, taking the latent state and base policy action as input. We train this residual actor via Dreamer against the learned reward model. Experiments were conducted on Robomimic tasks, and the numbers are reported in the table below (averaged with three seeds per task). We observe that SAILOR -- with test-time planning -- is better than learning a residual policy with backprop. We have added these results to the paper, demonstrating the importance of test-time planning.
>
> | | Lift (5 Demos) | Can (10 Demos) | Square (50 Demos) |
> |----------|----------|----------|----------|
> | DP |.58    | .33    | .28    |
> | WM + RM + Residual Policy|.85    | .61    | .54    |
> | SAILOR|.99    | .82    | .78    |
>
>
> > W1 (c): "How much real-world interaction does the WM save us?" Given the many differences between SAILOR and DPPO-IRL, it is hard to conclude if this is due to the WM or other algorithmic choices. For instance, DPPO-IRL uses PPO as the base optimizer, which is known to be very sample inefficient, while SAILOR is closer to an off-policy method as it reuses efficiently previous data. I wonder if other baselines can shed more light on this (e.g., can an off-policy diffusion-based approach like Diffusion Q-learning be adapted as an IRL variant?)
>
> We agree that adding an additional baseline where we use an off-policy RL algorithm as the policy optimizer for our model-free IRL baseline would strengthen our paper. However, we were not able to find a method that (a) trains diffusion policies, (b) handles image observations, and (c) has publicly released code, which made it a challenge to implement such a baseline within the rebuttal time frame. For example, none of the other algorithm implementations in the DPPO codebase work from images other than DPPO, and the off-policy RLPD codebase cannot directly train a diffusion model. We also extensively searched for an implementation of DQL and IDQL that worked from image observations and could not find one. We are happy to add this as a point to our limitations section.
>
> > W2: The claim "Our key insight is that we can infer the latent search process required to recover from local mistakes from the same source of human data required for the standard behavioral cloning pipeline" in the introduction also seems to contradict a bit the functionality of the algorithm: as SAILOR can rollout data in the environment, how can we conclude that it can recover from mistakes from expert data only?
>
> We thank the reviewer for highlighting this. In our original statement, we wanted to emphasize that SAILOR does not require any additional interaction/supervision from the *human* expert, just like behavioral cloning. This is in contrast to a method like DAgger [2]. We agree that saying we do not need any more "human data" is not as clear as we could have been, and have rephrased this in the paper.
>
> > W3: Section 3.1 mentions that "Beyond these benefits, the unique advantage of the L2S paradigm is a computational one: the ability to only need to plan at the states encountered at test-time". I was actually thinking the opposite, as planning methods are known to be computationally much more inefficient at test-time than pre-trained policies. I think one advantage is instead that planning implicitly expands the space of possible behaviors that can be generated (as opposed to a pre-trained policy which is limited to the capacity of a fixed neural net)
>
> We agree with the reviewer that in situations where computation is highly limited at inference time, it may not be possible to do multiple test-time evaluations. However, we found that for our SAILOR implementation, we only spent ~0.014s time per iteration on the MPPI residual search, compared to ~0.066s on the DP forward pass, a somewhat minimal overhead. This is likely because the MPPI planner operates on top of lower-dimensional *latent states*, rather than high-dimensional image observations like the base DP. We used the default parameters, which include 6 iterations of planning, and we empirically observed that the performance converges after 2-3 iterations with a marginal improvement in later iterations (as seen from the table below). We thank the reviewer for bringing this up, and have added this to the paper
>
> | Method                 | Lift (5 Demos) | Can (10 Demos) | Square (50 Demos)  |
> |-----|-----|----|----|
> |BaseDP|0.58|0.33|0.28|
> |1 Itr|0.99|0.78|0.61|
> |3 Itrs|0.99|0.83|0.72|
> |6 Itrs (SAILOR)|0.99|0.83|0.78|
> |9 Itrs|0.98|0.87|0.77|
>
> That said, in terms of total compute used for policy updates across training and test times (not just at training time), we believe that L2S methods are likely more efficient than traditional policies. Intuitively, this is because for a policy to be able to respond well to a scenario, it likely needs to have been trained on something similar at training time. In the worst case, we need to handle *all* such scenarios at train time, while an L2S method only needs to handle what is seen at test time. A more formal version of this argument can be found in the classic paper https://www.cis.upenn.edu/~mkearns/papers/sparsesampling-journal.pdf. Furthermore, **because we iteratively distill the output of the search process back into the base policy via the expert iteration subroutine, SAILOR avoids re-solving the same residual search problem in the future, essentially recycling the test-time compute**.
> Also, our ablation on learning a residual actor to predict actions shows that test-time planning outperforms learning with a residual actor-- validating the limited capacity with a fixed neural net as policy.
>
> > W4: The paper mentions data SAILOR can be applied on top of any base policy, but then the method is tested only on top of DPs. To really show this generality, I'd be better to add experiments on other base policies (eg recent VLAs as mentioned in the text)
>
> We agree that this is an interesting and promising direction for future work! In particular, we'd be interested in seeing how SAILOR could be applied to base VLA policies, and, similar to VLAs, can leverage text information as input while planning.
>
> We hope our response has addressed your concerns and resonated with you. We're happy to answer any further questions you may have and hope you consider increasing your score.
>
>
> Best Regards,\
> The Authors
>
> #### References
> [1] Ho et al., Generative Adversarial Imitation Learning, NeurIPS 2016.\
> [2] Ross et al., A Reduction of Imitation Learning and Structured Prediction to No-Regret Online Learning, AISTATS 2011.

---

> > ### Comment · Reviewer_FQJB · 2025-08-04
> >
> > Thanks for the very detailed answer! The additional experiments are great and I believe they do make the main conclusions convincing. I have raised my score accordingly.

---

### Official Review · Reviewer_AkFH · 2025-07-01

**Clarity:** 3
**Significance:** 2
**Originality:** 3
**Rating:** 4
**Confidence:** 3

**Summary:**

The paper presents SAILOR. Given nominal trajectories predicted from a diffusion policy, SAILOR tries to find a residual trajectory by searching under a world (RSSM) and a reward model (discriminator on “expert-like” behaviors).

**Questions:**

- I’d be interested to learn a variant of figure 6 where the x-axis is the total amount of compute used (EFLOPs).
- How does the proposed approach work on a real robot?

**Ethical Concerns:**

["NO or VERY MINOR ethics concerns only"]

**Final Justification:**

The responses to my questions regarding comparison to GPS (guided policy search) and Dreamer are fair. While I agree on that, it does not change the fact that despite not requiring ground-truth reward / dynamics, the proposed method only evaluates on simulation environments that do provide them, rather than on real-robots where the method is motivated on (not requiring ground-truth reward and dynamics). Therefore I am not fully convinced as the paper does not closed the "motivation-method-experiment" loop. However, given the paper's clarity and thorough experiments, and that the authors have addressed my other concerns, I want to raise my score to 4 at this point. I still strongly advise the authors to include real-robot experiments in the final version of the paper, if the paper were accepted.

**Limitations:**

Missing an explicit limitation discussion section. It's unclear how the proposed approach would work on a real robots.

**Quality:**

3

**Strengths And Weaknesses:**

### Strengths
+ The paper is well-motivated and presented
+ Detailed ablation analysis and I appreciate the clarity in disclosing the key findings in tuning the proposed method (section 4.2)

### Weaknesses
- My main concern is the evaluation setup. The main metrics measured is success rate yet the core idea is to improve BC. Success rate, especially in simulators, can be easily improved using RL without using world model or reward model.
- Related to above, please also consider comparing to other closed-loop IL methods, such as GAIL (that also trains the reward model on-the-fly).
- The proposed method on a high-level looks very much like Dreamer, especially given the RSSM world model, yet did not ablate the core difference against Dreamer which is the local search. How important is the search compared to backprop through model in Dreamer?
- Missing an explicit limitation discussion section.
- If possible, I'd like to see a comparison on data/compute-efficiency against GPS (guided policy search).

---

> ### Author Rebuttal · Authors · 2025-07-31
>
> Thank you for your valuable comments that helped improve the paper. We are delighted you found our analysis detailed and appreciated the key findings of SAILOR in our ablations. We respond to your questions below:
>
> > W1: My main concern is the evaluation setup. The main metrics measured is success rate yet the core idea is to improve BC. Success rate, especially in simulators, can be easily improved using RL without using world model or reward model.
>
> We would like to clarify that we do not assume access to the ground truth reward function (e.g., success rate), which means we cannot directly use RL, as there is no reward function to run it with. More generally, **specifying reward functions (i.e., solving a *reward design problem*, [1]) is often quite challenging in practice**. For example, in domains like self-driving, it is hard to hand-engineer a reward function that captures what good driving is -- one would need to write down the precise trade-offs between getting an inch closer to another car and arriving at the destination a few minutes earlier. Instead, in inverse RL techniques like SAILOR, one *infers* such a reward function from demonstrations of good behavior, side-stepping this problem.
>
> > W2: Related to above, please also consider comparing to other closed-loop IL methods, such as GAIL (that also trains the reward model on-the-fly).
>
> Our DPPO-IRL baseline is a stronger version of traditional GAIL that optimizes over diffusion policies. The policy is optimized via (D)PPO as in GAIL. The discriminator/reward model is trained iteratively as in GAIL. We show that across multiple tasks, SAILOR is significantly more interaction-efficient than DPPO-IRL.
>
> > W3: The proposed method on a high-level looks very much like Dreamer, especially given the RSSM world model, yet did not ablate the core difference against Dreamer which is the local search. How important is the search compared to backprop through model in Dreamer?
>
> We would like to again clarify that we do not assume access to a ground-truth reward function and therefore cannot run vanilla Dreamer, as there is no reward signal. That said, once we have inferred a reward model via training a classifier between expert demonstrations and learner rollouts as in inverse RL (e.g. GAIL), it is an interesting question whether residual search using the learned reward/world model performs better than learning a *residual policy* via backpropping through the learned world model -- we thank the reviewer for suggesting this ablation! More explicitly, the residual actor network is tasked with predicting a corrective action, taking as input the latent state and base policy action. We trained this policy in standard DREAMER fashion (i.e., via backprop through the world model). Our experiments in the table below (averaged with 3 seeds per task) show that **SAILOR with test-time planning is better than learning a residual actor**. We have added these results to our paper. We have added these results to the paper, where this study demonstrates the importance of test-time planning.
>
> | | Lift (5 Demos) | Can (10 Demos) | Square (50 Demos) |
> |----------|----------|----------|----------|
> | DP |.58    | .33    | .28    |
> | SAILOR (Residual Policy) |.85    | .61    | .54    |
> | SAILOR|.99    | .82    | .78    |
>
> > W4: Missing an explicit limitation discussion section.
>
> We agree with the reviewer that adding an explicit discussion and limitation section would further strengthen our manuscript. We have added the following and are happy to move it to the additional page for camera-ready or the appendix:
>
> ### Limitations
> At a high level, we believe there are two key limitations in our work that we believe offer promising opportunities for future work. The first is to perform experiments on real robots -- recent work FOREWARN [2] has shown that similar ideas to SAILOR can be scaled to real robots via the combination of diffusion policies, the Dreamer WM architecture, using VLMs as reward functions, and test-time planning. We believe that adding in other components of the SAILOR architecture (e.g., the use of a residual search rather than mode selection as in FOREWARN) could further improve performance. Second, we currently train the world model from scratch. Instead, leveraging a pre-trained foundation world model like Dino WM or V-JEPA could further improve sample efficiency and potentially eliminate the need for a warm start.
>
> ### Discussions
> In summary, we see that across a dozen visual manipulation tasks, SAILOR is able to train robust agents that recover from the failures of the base policy via test-time planning and match the expert’s outcomes, all without any additional human data. Furthermore, scaling up the amount of data used to train a diffusion policy via behavioral cloning by ≈ 5-10× is unable to close this performance gap. We also see that our learned RM is able to detect nuanced failures, while the SAILOR agent is able to recover from them. As mentioned above, while for our experiments we use a base diffusion policy, MPPI planner, and Dreamer world model, the general SAILOR architecture is fully compatible with foundation behavior (e.g., VLAs) and world models trained on internet-scale data, which might unlock further levels of generalization and robustness – we leave this as a promising future direction.
>
> > If possible, I'd like to see a comparison on data/compute-efficiency against GPS (guided policy search).
>
> If by GPS, the reviewer is referring to [3] (an excellent paper we are huge fan of), we would like to note both that (a) we cannot directly apply the original GPS from image observations and (b) SAILOR can already be thought of as a generalization of many of the ideas in GPS. Ignoring some of the lower-level details, GPS essentially boils down to using an iLQR/DDP planner as a local expert that gets distilled into a policy via expert iteration. This requires access to a low-dimensional state to apply iLQR/DDP (we operate from images) and assumes dynamics are well-approximated by piecewise linear functions (we make no such assumptions, which likely do not hold in image-space). However, via its expert iteration subroutine, SAILOR periodically distills the output of the search process into the base policy. In this sense, **SAILOR can be seen as a modern generalization of GPS that uses Dreamer to handle image observations, MPPI rather than iLQR / DDP to handle non-smooth dynamics, and does not require access to a ground-truth reward function**. We have updated our related work section to make this point more explicit.
>
> > Q1: I’d be interested to learn a variant of figure 6 where the x-axis is the total amount of compute used (EFLOPs).
>
> We thank the reviewer for the suggestion. We have computed the total amount of compute for our model in GFLOPs and have included these numbers in the revised plot in our paper.
>
> **Base Policy:** 14.554 GFLOPs\
> **MPPI Planner:** 0.1637 x `num_itrs` x `num_samples` GFLOPs
>
> We would like to point out that since the MPPI planner can process all samples in parallel, the inference time only depends on `num_itrs` and is 0.014s per MPPI iteration (and 0.066s for the base diffusion policy). In the paper, we compute numbers with 256 samples and 6 iterations, but we observed that there are marginal gains after 2-3 iterations.
>
> > Q2: How does the proposed approach work on a real robot?
>
> In this work, we conduct experiments in simulation. FOREWARN [2] deployed DP with test-time planning with Dreamer world model and VLM on a real robot, and we believe similar setups can be utilized to deploy SAILOR on real hardware. We thank the reviewer for pointing this out and have added this in the limitations of our work.
>
> We hope our responses and proposed changes have resonated with you, and you consider increasing your score.
>
> Best regards,\
> The Authors
>
> #### References
> [1] Hadfield-Menell, Inverse Reward Design, NeurIPS 2025.\
> [2]  Wu et al., From Foresight to Forethought: VLM-In-the-Loop Policy Steering via Latent Alignment, RSS 2025.\
> [3] Levine et al., Guided Policy Search, ICML 2013.

---

> ### Comment · Reviewer_AkFH · 2025-08-03
>
> Thank you for the detailed response. The responses to my questions regarding comparison to GPS (guided policy search) and Dreamer are fair. While I agree on that, it does not change the fact that despite not requiring ground-truth reward / dynamics, the proposed method only evaluates on simulation environments that **do** provide them, rather than on real-robots where the method is motivated on (not requiring ground-truth reward and dynamics). Therefore I am not fully convinced as the paper does not closed the "motivation-method-experiment" loop. However, given the paper's clarity and thorough experiments, and that the authors have addressed my other concerns, I want to raise my score to 4 at this point. I still strongly advise the authors to include real-robot experiments in the final version of the paper, if the paper were accepted.

---

### Official Review · Reviewer_dq9G · 2025-07-02

**Clarity:** 4
**Significance:** 3
**Originality:** 3
**Rating:** 5
**Confidence:** 4

**Summary:**

This paper introduces SAILOR, an online framework for learning an imitation policy with robust recovery. SAILOR learns a world model, reward models, and a policy on mixed online interaction and offline expert demonstration data. SAILOR first trains a base BC policy on expert data, and populate the buffer with online rollouts. Then, it pretrains the world model and reward models on the mixed buffer. The reward models (RM, V) are trained to discriminate between expert and base policy rollouts. Then, during inference, SAILOR searches in world model latent space and optimizes the trajectory to match the expert demonstrations by maximizing the predicted rewards, and rolls out with receding horizon MPC. Results in various environments show that SAILOR is more robust than vanilla DP and is more efficient than inverse RL. Ablations validate the necessity of each component in SAILOR.

**Questions:**

- The paper mentioned that 20\% of the budget is allocated for warm-start. For each task, how many frames of online vs. offline trajectories do you have? How many frames minimum does it require to yield improved WM/RM/policy learning afterwards? It would be great to see a sweep of warm-start budget from 0\% to 20\% and its impact on downstream performance.
- For the distillation step, is it doing straightforward BC training with the new data? Do you have any data quality assumption on the collected online trajectories (e.g. suboptimal trajectories, success/failure assumptions)?
- In Figure 4, why do you think vanilla DP should plateau, given the same interaction budget? How do the SAILOR rollouts differ from the expert demos? Is it a data coverage/diversity problem?
- Reference links didn't seem to work.

**Ethical Concerns:**

["NO or VERY MINOR ethics concerns only"]

**Final Justification:**

I think SAILOR proposes a valid and novel idea for robust imitation learning, given the extensive evaluations on simulated environments, as well as related works on real robots exploring similar directions. I maintain my overall rating of accept.

**Limitations:**

- No real robot experiments.

**Paper Formatting Concerns:**

N/A.

**Quality:**

4

**Strengths And Weaknesses:**

- Robust recovery from out-of-distribution states for imitation learning is an important problem.
- The proposed method is well-motivated and novel. The authors introduce a model-based learning/planning approach for learning a robust imitation policy from expert demos and additional online interactions.
- Experiment results in various sim visual manipulation environments show that SAILOR is more robust than vanilla DP, and more efficient than inverse RL baselines. The reward model learns interpretable rewards for the task.
- Extensive ablations validate the necessity of each component in SAILOR.
- No real robot experiments.

---

> ### Author Rebuttal · Authors · 2025-07-31
>
> Thank you for your valuable feedback and questions. We are glad that you appreciate the robust recovery at OOD states, and the Interpretability of learned rewards of SAILOR. We respond to your questions below:
>
> > Q1: The paper mentioned that 20% of the budget is allocated for warm-start. For each task, how many frames of online vs. offline trajectories do you have? How many frames minimum does it require to yield improved WM/RM/policy learning afterwards? It would be great to see a sweep of warm-start budget from 0% to 20% and its impact on downstream performance.
>
> The number of frames of offline/expert data depends both on the number of expert demonstrations provided, as well as the length of each episode (100-250, depending on the task). The number of frames of online data is equal to the number of environment interactions -- we provide these numbers in Table 2 in the Appendix in the supplementary material. The warm-start percentage of 20% refers to the percentage of total interaction budget we use to collect data with the base policy to warm-start the WM and Critic. As requested, we performed an ablation of this percentage on the Square task with 50 demonstrations. We train agents with {0, 10, 20, 30}% of interaction dedicated to the warm-start. In the table below, **we observe that using 10% the interaction budget for a warm-start improves performance over no warm-start (performance measured across 3 seeds). However, performance plateaus after using 20% for the warm-start**. Ideally, we would want to allocate most of the interaction budget to online collection and policy update; hence, we used 20% for our experiments. We have added this ablation to the paper.
>
> | | Square (50 demos) |
> |----------|----------|
> |DP | .28 |
> | No Warmstart | .54    |
> | Warmstart (10%) |  .72  |
> | Warmstart (20%) |  .78  |
> | Warmstart (30%) |  .79  |
>
> > Q2: For the distillation step, is it doing straightforward BC training with the new data? Do you have any data quality assumptions on the collected online trajectories (e.g., suboptimal trajectories, success/failure assumptions)?
>
> Yes! We are literally doing behavioral cloning (BC) for the policy update, which makes our expert iteration procedure simple and scalable. More explicitly, as outlined in Algorithm 2, after sampling some observations from the agent's replay buffer, we relabel these with actions generated by the current SAILOR stack, and use this as the dataset to update the base policy. **There are no data quality assumptions as we do not have access to any environment rewards or success flags**.
>
> > Q3: In Figure 4, why do you think vanilla DP should plateau, given the same interaction budget? How do the SAILOR rollouts differ from the expert demos? Is it a data coverage/diversity problem?
>
> To clarify, in Figure 4, the x-axis represents the number of expert demonstrations provided to all methods, rather than the interaction budget -- the base DP does not have any interaction budget as it is trained via purely offline behavioral cloning. That said, the reviewer asks two interesting questions. The first is **whether BC can plateau at suboptimal performance even when provided with ample expert demonstrations**. The answer to this question is a **resounding yes** when the learner cannot perfectly imitate the expert -- i.e., the "misspecified setting" [1]. In such settings, mistakes are inevitable, and the goal is to avoid particularly bad mistakes that compound over the horizon (e.g., falling off a cliff) [2]. However, without the ability to observe the consequences of one's actions, a purely offline algorithm is unable to distinguish between mistakes that don't matter (e.g., an unnecessary lane change) and those that do (e.g., hitting another car) [3]. In such settings, the hope is that an interactive imitation learning algorithm like SAILOR is able to learn to correct from recoverable mistakes back to the expert's path. The second question the reviewer asks is also quite insightful: **If the expert demonstrations contained the desired recovery behavior, is there a need for interaction?** If, for example, one is able to pre-hoc predict *all* mistakes *any* policy could make, one could include this in the demonstration set, but this is, of course, a tall order. In general, this can require an amount of recovery demonstrations that scales *exponentially* with the task horizon. Instead, however, one can just observe what mistakes a particular chosen policy makes, and **can either have a human show how to recover from these mistakes (as in DAgger) or have the learner figure this out without further human guidance (as in SAILOR)**.
>
> > Q4: Reference links didn't seem to work.
>
> Ah, thank you for the catch! We had a Chrome extension to cut the PDF for submission that hid this fact from us. The links are working in the paper submitted with the Appendix in the supplementary material, and we will fix this in the revised version of the paper.
>
> > Q5: No real robot experiments.
>
> We agree with the reviewer that real robot experiments would strengthen our story and leave it as a promising direction for future work (and one that we are currently working on!). However, the reviewer might find it interesting that the just-published FOREWARN paper [4] showed that a similar setup to SAILOR can be scaled to real robots.
>
> We hope we addressed most of your concerns, and hope you consider increasing your score.
>
> Best regards,\
> The Authors
>
> #### References
> [1] Espinosa-Dice et al., Efficient Imitation under Misspecification, ICLR 2025.\
> [2] Ross et al., A Reduction of Imitation Learning and Structured Prediction to No-Regret Online Learning, AISTATS 2011.\
> [3] Swamy et al., Of Moments and Matching: A Game-Theoretic Framework for Closing the Imitation Gap, ICML 2021.\
> [4] Wu et al., From Foresight to Forethought: VLM-In-the-Loop Policy Steering via Latent Alignment, RSS 2025.

---

> > ### Comment · Reviewer_dq9G · 2025-08-04
> > **Thank you for the clarifications and additional experiments**
> >
> > Thank you for the detailed rebuttal reply. Overall I am positive about this paper and I think SAILOR proposes a valid and novel idea for robust imitation learning, given the extensive evaluations on simulated environments, as well as related works on real robots exploring similar directions.

---

### Official Review · Reviewer_yCh2 · 2025-07-03

**Clarity:** 4
**Significance:** 3
**Originality:** 4
**Rating:** 5
**Confidence:** 4

**Summary:**

SAILOR augments an off-the-shelf behaviour-cloned diffusion policy with a learned world model, reward discriminator and latent-space MPPI planner. At test time it treats the base policy’s $k$-step plan as a proposal, samples residual trajectories in the world model, scores them with the learned reward model and critic, and executes the first action of the best-scoring plan. Across 12 simulated manipulation tasks and three data scales, SAILOR consistently beats the underlying DP, even when DP is given 5–10 × more demonstrations, and does so with far fewer real-environment interactions. The paper backs the design with three key ablations, demonstrates that SAILOR can directly correct its base policy, and shows that planning remains stable when the search budget is pushed.

**Questions:**

1. In Figure 2, the reward model (RM) is also used to evaluate the trajectories generated by DP. However, DP does not incorporate such a component. Where does the RM used to evaluate DP originate? Is it the same RM trained for SAILOR?
2. Why are different tasks used in Figure 9 when ablating warm-start compared to the other components? Would all tasks confirm your findings?
3. In the hybrid WM ablation (Fig. 9, center), does “no hybrid” mean using only expert demos or only on-policy rollouts?

**Ethical Concerns:**

["NO or VERY MINOR ethics concerns only"]

**Final Justification:**

In light of the clarifications to my comments (W2: on-policy training, W3: compute time, and W5: claim about offline data), the extra distillation results (W4), and responses to other reviewers, I have decided to **increase the score**.

**Limitations:**

No potential negative societal impact. I commend the authors for boldly marking “N/A” in the checklist regarding the limitations of their work. However, some practical constraints and assumptions, such as reliance on a competent base policy and computational demands of test-time planning, as discussed above, to name a few, deserve acknowledgment.

**Paper Formatting Concerns:**

No concerns.

**Quality:**

3

**Strengths And Weaknesses:**

# Strengths

1. With the same number of expert demonstrations, SAILOR outperforms DP on a wide array of *12* tasks from different robotic arm suites, and DP can’t catch up even with 10× data. SAILOR also requires far fewer interactions than a model-free IRL baseline (DPPO-IRL). Its benefits extend beyond the typical few-shot imitation learning use case.
2. Figures 2 and 7 effectively illustrate SAILOR’s ability to recover from base policy errors. The visualizations combine scenario snapshots with reward model curves, showing clear course-correction behavior during execution. Moreover, consistent color coding across all figures enhances readability and makes comparisons between SAILOR and DP intuitive.
3. The method only requires a nominal plan from any pretrained policy along with an image observation, making it agnostic to the underlying architecture. Nothing ties it specifically to diffusion models, suggesting a modular, plug-and-play design that could generalize to a broader range of base policies.
4. The paper is very well written and easy to follow. Design choices are consistently justified with logical reasoning, clear explanations, and relevant citations. It is well-situated within the literature on imitation learning, planning, L2S, and reward modeling, with strong connections drawn to prior work throughout.
5. The ablation study sharpens the case for the chosen model/method components. By isolating warm-starting, hybrid world model updates, and expert iteration, the authors show that each component significantly affects performance.
6. As demonstrated in the ablation study, the expert iteration loop enables SAILOR to progressively refine the base policy, reducing reliance on costly test-time planning in recurring scenarios.

# Weaknesses

1. The paper treats L2S as a universally appealing paradigm due to its training-time planning efficiency. However, this ignores scenarios where inference-time compute is limited, yet ample compute is available during training. In such settings, SAILOR's reliance on expensive MPPI rollouts at every timestep may be impractical, and more traditional offline methods or distilled policies would be preferable. While the authors briefly mention expert iteration for distillation, they don’t evaluate or emphasize it as a viable path for deployment, leaving a gap between their method's design and real-world utility.
2. According to the ablation study, warm-starting plays a critical role in overall performance. To facilitate this, SAILOR assumes the base policy is sufficiently competent to provide useful warm-starts for learning a good prior. If the base policy performs poorly, the initial rollouts used to train the world model and reward model will be low-quality, leading to inaccurate dynamics and reward predictions. This, in turn, undermines the effectiveness of MPPI planning.
3. While SAILOR clearly outperforms DP, this isn’t the fairest comparison. DP is a purely offline, feedforward policy with no planning, reward signal, or test-time adaptation, whereas SAILOR leverages a learned world model, reward model, and expensive online planning. The absence of any measurement of inference latency, MPPI rollout time, or runtime comparisons with DP or DPPO-IRL is a major omission, especially given SAILOR’s heavy test-time compute. DPPO-IRL serves as a more appropriate baseline, as it also learns a reward model and improves the policy beyond BC. However, it shifts the compute burden to training via model-free RL, whereas SAILOR achieves higher performance with significantly fewer environment interactions. A more meaningful comparison would account for total compute, both in terms of sample efficiency and wall-clock time.
4. While the framework is presented as modular and capable of operating with any arbitrary base policy capable of generating a sequence of future actions, in practice, the paper only demonstrates integration with DP. To substantiate this claim, the authors could empirically evaluate SAILOR with alternative base policies.
5. The authors state that as a key insight “*we can infer the latent search process required to recover from local mistakes from the same source of human data required for the standard behavioral cloning pipeline.*” While SAILOR does leverage expert demonstrations, this claim overstates their centrality. The core training of the world model, reward model, and critic relies heavily on **on-policy rollouts** from the base policy and the SAILOR stack. Expert data is used, but not exclusively, and much of the method’s performance stems from its ability to adapt online. The implication that SAILOR learns solely from offline BC-style data is misleading.

# Minor Suggestions

1. It took me a while to figure out that the dots on the line plots in Figure 2 (Right) represent when the image snapshots above were taken. Perhaps mention this in the caption for clarity.
2. If I understand correctly, the x-axis values in Figure 8 should be powers of two, instead of values represented with E notation.

---

> ### Author Rebuttal · Authors · 2025-07-31
>
> Thank you for your valuable review of our paper. We are pleased to hear that you found the paper easy to understand and appreciated SAILOR’s ability to recover from base policy errors with fewer environment interactions. We respond to questions below.
>
> > W1: While the authors briefly mention expert iteration for distillation, they don’t evaluate or emphasize it as a viable path for deployment.
>
> In terms of total compute used for policy updates across training and test times (not just at training time), we believe that L2S methods are likely more efficient than traditional policies. Intuitively, this is because for a policy to be able to respond well to a scenario, it likely needs to have been trained on something similar at training time. **In the worst case, we need to handle *all* such scenarios at train time, while an L2S method only needs to handle what is seen at test time.** A more formal version of this argument can be found in the classic paper https://www.cis.upenn.edu/~mkearns/papers/sparsesampling-journal.pdf. Furthermore, because we iteratively distill the output of the search process back into the base policy via the expert iteration subroutine, SAILOR avoids re-solving the same residual search problem in the future, essentially recycling the test-time compute.
>
> That said, we agree with the reviewer that in situations where computation is highly limited at inference time, it may not be possible to successfully employ an L2S strategy. However, we found that for our SAILOR implementation, **we only spent ~0.014s time per iteration on the MPPI residual search, compared to ~0.066s on the DP forward pass** (we used 6 iterations of MPPI in reported results). This is likely because the MPPI planner operates on top of lower-dimensional *latent states*. In addition, we saw that performance converges after 2-3 iterations, and there is only a marginal improvement in later iterations.
>
> Inspired by the reviewer's question, **we explored an additional option: performing a post-hoc final distillation of the SAILOR agent to base policy**. This helps the base policy to close the gap in performance with the search policy (shown in the table below). As this base policy will be as fast as DP at test time, it is suitable for deployment. In addition, the test-time planner can provide additional gains with more inference budget, and we can see that the planner adds marginal gains after 3 iterations. Lastly, we would like to highlight that MPPI planners for test-time planning have been extensively deployed on real robots, including the work of [1,2,3,4], implying that it is not a barrier to deployment. We thank the reviewer for suggesting this experiment and have added it to the paper.
>
> |        | Lift (5 Demos) | Can (10 Demos) | Square (50 Demos) | Inference Time (s)  |
> |--------|----------------|----------------|-------------------|----------------|
> | DP (pre-training)   | 0.58           | 0.33           |  0.28             | 0.066             |
> | DP (post-trainng)  | 0.94 |0.81 | 0.56 | 0.066 |
> | + MPPI 1 Iter | 0.98 | 0.84           |  0.71             | 0.08          |
> | + MPPI 3 Iter | 0.99 | 0.84          |  0.8             | 0.109          |
> | + MPPI 6 Iter | 0.99           | 0.85           |  0.8             | 0.151          |
> | + MPPI 9 Iter | 0.98           | 0.85           |  0.81             | 0.195          |
>
> > W2: If the base policy performs poorly, the initial rollouts used to train the world model and reward model will be low-quality, leading to inaccurate dynamics and reward predictions.
>
> We agree with the reviewer that when the base policy is weaker, the learner is fundamentally faced with solving a harder search problem. However, **because both the world and reward model are trained on on-policy data, we should still be able to improve upon the base policy via search**. In some sense, the "quality" of a world/reward model is perhaps better measured by how useful they are for improving the current policy, rather than how accurate they are on a good policy's distribution -- see [5] for more information. That said, we do see at least some empirical evidence that SAILOR is able to achieve a high success rate even when the base policy does not. For example, for Door (1 demos) and PokeCube (5 demos), the pretrained DP has 0.09 and 0.2 SR, and SAILOR achieves 1.0 and .91 SR after training.
>
> > W3: The absence of any measurement of inference latency, MPPI rollout time, or runtime comparisons with DP or DPPO-IRL is a major omission.
>
> We thank the reviewer for the suggestions. First, we would like to emphasize that no method in our paper (including SAILOR) gets access to any sort of "ground-truth" reward signal -- both DPPO-IRL and SAILOR infer a reward model from demonstrations rather than, say, regressing ground-truth reward labels. That said, we agree with the reviewer that including more information about the wall-clock time each of these methods takes at test-time would significantly strengthen our paper. As mentioned above, we only spent ~0.014s per iteration on the MPPI residual search, compared to ~0.066s on the DP forward pass (which all methods pay for).
>
> In terms of wall-clock training time, SAILOR takes around 40 hours and DPPO-IRL takes around 11 hours for 500K environment steps on a single GPU (NVIDIA 6000Ada with 48 GB memory). We attribute this higher train time to the computation required to train a reconstruction-based world model like DREAMER. However, in terms of environment interactions required to reach a particular performance, we robustly find SAILOR requires significantly fewer than DPPO-IRL, as our results in the paper provide evidence for. Furthermore, if instead we measure the model performance given a fixed wall-clock time, we often find SAILOR to be significantly better than DPPO-IRL. For example, **on the Door task, SAILOR converges to a success rate of 1.0 in ~11 hrs with about 100k environment steps, while DPPO-IRL reaches a success rate of just 0.75 (with 500k interactions and ~11 hrs of training)**. We have included these numbers in the paper as they paint a more nuanced picture of the compute requirements for different methods across training and testing. Please let us know if there are any other comparisons you think would strengthen our story.
>
> > W4:  To substantiate this claim, the authors could empirically evaluate SAILOR with alternative base policies.
>
> We agree that this is an interesting and promising direction for future work! In particular, we'd be interested in seeing how SAILOR could be applied to base VLA policies.
>
> > W5: The implication that SAILOR learns solely from offline BC-style data is misleading.
>
> We thank the reviewer for highlighting this. In our original statement, we wanted to emphasize that SAILOR does not require any additional interaction/supervision from the *human* expert, just like behavioral cloning. This is in contrast to a method like DAgger [6]. We agree that saying we do not need any more "human data" is not as clear as we could have been, and have rephrased this in the paper.
>
> > Q1: Where does the RM used to evaluate DP originate? Is it the same RM trained for SAILOR?
>
> Yes, the RM trained with SAILOR was used for the evaluation of all policies. In particular, we first generate rollouts using the DP and score them with the reward model, then reset to a state on this trajectory to rollout from with SAILOR. This shows that the SAILOR RM is able to distinguish between successful and unsuccessful rollouts and identify nuanced failures in the base policy. We have added a note on this to the caption of Figure 2.
>
> > Q2: Why are different tasks used in Figure 9 when ablating warm-start compared to the other components? Would all tasks confirm your findings?
>
> In our experiments, different tasks showed different amounts of dependence on each component of the overall SAILOR stack. That said, all components show a significant impact on at least a small subset of tasks, as the plots confirm. Thus, to have a method that performs uniformly well across all tasks, we believe we need the full SAILOR stack. We have added this point to the caption in the paper.
>
> > Q3: In the hybrid WM ablation (Fig. 9, center), does “no hybrid” mean using only expert demos or only on-policy rollouts?
>
> In the Fig.9, no hybrid means using only on-policy rollouts. We have updated the description of Fig. 9 to reflect this point.
>
> > Minor suggestions
>
> This is a great catch. We have revised the caption of Figure 2 and updated Figure 9 to change E notation to powers of two.
>
> > Limitations
>
> We have added a discussion and limitations section to the revised version. We discuss the lack of real-robot experiments and that learning from scratch often requires too many interactions to be feasible in practice, potentially incentivizing the use of foundation models for base policies and world models. Due to space constraints, we refer the reviewer to our response to Reviewer kaSH (W3) in the rebuttal for further details.
>
> Thank you for the thoughtful comments. We hope we addressed most of your concerns, and hope you consider increasing your score.
>
> Best regards,\
> The Authors
>
> ### References
> [1] Minarik et al., Model Predictive Path Integral Control for Agile Unmanned Aerial Vehicles, 2024\
> [2] Testouri et al., Towards a Safe Real-Time Motion Planning Framework for Autonomous Driving Systems: A Model Predictive Path Integral Approach, 2024.\
> [3] Han et al., Model Predictive Control for Aggressive Driving Over Uneven Terrain, RSS 2024.\
> [4] Lancaster et al., MoDem-V2: Visuo-Motor World Models for Real-World Robot Manipulation, 2024.\
> [5] Ross et al., Agnostic System Identification for Model-Based Reinforcement Learning, ICML 2012.\
> [6] Ross et al, A Reduction of Imitation Learning and Structured Prediction to No-Regret Online Learning, AISTATS 2011.

---

> > ### Comment · Reviewer_yCh2 · 2025-08-03
> >
> > In light of the clarifications to my comments (W2: on-policy training, W3: compute time, and W5: claim about offline data), the extra distillation results (W4), and responses to other reviewers, I have decided to **increase the score**. Thank you for your work!

---

### Decision · Program_Chairs · 2025-09-17

**Decision:**

Accept (spotlight)

**Comment:**

The paper addresses the the key limitation of Behaviour Cloning: lack of ability to recover from unseen states by extending BC with three blocks (namely a world model, a reward model and a latent-space planner) and a training regime (including ideas from data aggregation and policy distillation) that shifts the paradigm from imitation learning towards learning to search. The architecture is also modular and agnostic to the policy architecture. Simulation results show that the proposed approach outperforms vanilla BC, even when BC is trained with 5-10 times the data, demonstrating that it can robustly recover from policy errors, and it is also more sample efficient than a model-free inverse-RL baseline. The paper is also clear and well written.

Reviewers appreciated all these contributions and expressed concerns about the evaluation, applicability and assumptions. The authors addressed these comments with the rebuttal and during the discussion period, running experiments with a new baseline and clarifying multiple points satisfactorily. They also acknowledge the limitation of claiming the method agnostic on the policy while it has been trained on diffusion BC only, and agreed with the reviewers that trying with VLAs would be an interesting future work.

I recommend acceptance as a spotlight because the contributions are significant, the idea is novel and sound, and the paper is clear and well-written. Overall, it is a very promising direction in imitation learning that I believe will inspire further research.